# Unsupervised deep learning identifies semantic disentanglement in single inferotemporal face patch neurons

Irina Higgins [1,7 ✉], Le Chang[2,3,7], Victoria Langston[1], Demis Hassabis [1,4], Christopher Summerfield[1,5], Doris Tsao [2,6] & Matthew Botvinick [1,4]

In order to better understand how the brain perceives faces, it is important to know what objective drives learning in the ventral visual stream. To answer this question, we model neural responses to faces in the macaque inferotemporal (IT) cortex with a deep self-supervised generative model, $\beta$-VAE, which disentangles sensory data into interpretable latent factors, such as gender or age. Our results demonstrate a strong correspondence between the generative factors discovered by $\beta$-VAE and those coded by single IT neurons, beyond that found for the baselines, including the handcrafted state-of-the-art model of face perception, the Active Appearance Model, and deep classifiers. Moreover, $\beta$-VAE is able to reconstruct novel face images using signals from just a handful of cells. Together our results imply that optimising the disentangling objective leads to representations that closely resemble those in the IT at the single unit level. This points at disentangling as a plausible learning objective for the visual brain.

[1] DeepMind, London, UK. [2] Caltech, Pasadena, USA. [3] Institute of Neuroscience, CAS Center for Excellence in Brain Science and Intelligence Technology, Chinese Academy of Sciences, Shanghai, China. [4] University College London, London, UK. [5] University of Oxford, Oxford, UK. [6] Howard Hughes Medical Institute, Pasadena, USA. [7] These authors contributed equally: Irina Higgins, Le Chang. ✉email: irinah@google.com

It is well known that neurons in the ventral visual stream support the perception of faces and objects[1]. Decades of extracellular single neuron recordings have defined their canonical coding principles at different stages of the processing hierarchy, such as the sensitivity of early visual neurons to oriented contours and more anterior ventral stream neurons to complex objects and faces[2,3]. A sub-network of the inferotemporal (IT) cortex specialised for face processing is particularly well studied[3–5]. Faces appear to be represented within such patches using low-dimensional neural codes, where each neuron encodes an orthogonal axis of variation in the face space[3]. An important yet unanswered question is how such representations may arise through learning from the statistics of the visual input. The most successful computational model of face processing, the active appearance model (AAM)[6], is a largely handcrafted framework which cannot help answer this question. Can we find a general learning principle that could match AAM in terms of its explanatory power, while having the potential to generalise beyond faces?

Recently, deep neural networks have emerged as popular models of computation in the primate ventral stream[7,8]. Unlike AAM, these models are not limited to the domain of faces, and they develop their tuning distributions through data-driven learning. Such contemporary deep networks are trained with high-density teaching signals on multiway object recognition tasks[9], and in doing so form high-dimensional representations that, at the population level, closely resemble those in biological systems[10–12]. Such deep classifiers, however, currently do not explain the responses of single neurons in the primate face patch better than AAM[6]. Furthermore, deep classifiers and AAM differ in their representational form. While deep classifiers develop high-dimensional representations where information is multiplexed over many simulated neurons, AAM has a low-dimensional code where single dimensions encode orthogonal information. Hence, the question of whether there exists a learning objective that leverages the power of deep neural networks while preserving the "gold standard" representational form and explanatory power of the handcrafted AAM remains open.

An important further challenge for theories that rely on deep supervised networks is that external teaching signals are scarce in the natural world, and visual development relies heavily on untutored statistical learning[13–15]. Building on this intuition, one longstanding hypothesis[1,16] is that the visual system uses self-supervision to recover the semantically interpretable latent structure of sensory signals, such as the shape or size of an object, or the gender or age of a face image. While appearing deceptively simple and intuitive to humans, such interpretable structure has proven hard to recover in practice, since it forms a highly complex non-linear transformation of pixel-level inputs. Recent advances in machine learning, however, have offered an implementational blueprint for this theory with the advent of deep self-supervised generative models that learn to "disentangle" high-dimensional sensory signals into meaningful factors of variation. One such model, known as the beta-variational autoencoder (β-VAE), learns to faithfully reconstruct sensory data from a low-dimensional embedding whilst being additionally regularised in a way that encourages individual network units to code for semantically meaningful variables, such as the colour of an object, the gender of a face, or the arrangement of a scene (Fig. 1a–c)[17–19]. These deep generative models thus continue the longstanding tradition from the neuroscience community of building self-supervised models of vision[20,21], while moving in a new direction that allows strong generalisation, imagination, abstract reasoning, compositional inference and other hallmarks of biological visual cognition[18,22–24].

In this work we compare the responses of single neurons in the primate IT face patches and single units learnt by different computational models when presented with the same face images. Our goal is to answer the question of whether a general learning objective can give rise to an encoding that matches the representational form employed by the real neurons. This question has so far been ignored in the literature, with most quantitative results instead reporting measures of explanatory power[3,6,8,11,25–28] that are insensitive to the representational form[15]. Our results demonstrate that the disentangling objective optimised by β-VAE is a viable option for explaining how the ventral visual stream develops the observed low-dimensional face representations[3]. We find significantly stronger one-to-one correspondence between the responses of single units learnt by β-VAE and the responses of single IT neurons compared to all other baselines, including deep classifiers and AAM. β-VAE also produces more accurate reconstructions of novel faces than the alternative methods when decoding from the activity of a handful of face-patch neurons. Furthermore, β-VAE learns using a general self-supervised objective without relying on high-density teaching signals like deep classifiers, which makes it more biologically plausible.

## Results

**Single disentangled units explain the activity of single neurons.** If the computations employed in biological sensory systems resemble those employed by this class of deep generative model to disentangle the visual world, then the tuning properties of single neurons should map readily onto the meaningful latent units discovered by the β-VAE. Here, we tested this hypothesis, drawing on a previously published dataset[6] of neural recordings from 159 neurons in macaque face area AM, made whilst the animals viewed 2100 natural face images (Fig. 2a, see "Methods"). We first investigated whether the variation in average spike rates of any of the individual recorded neurons was explained by the activity in single units of a trained β-VAE that learnt to "disentangle" the same face dataset that was presented to the primates. For illustration, in Fig. 1c we show faces that were generated (or "imagined") by such a β-VAE. Each row of faces is produced by gradually varying the output of a single network unit (we call these "latent units"), and it can be seen that they learnt to encode fairly interpretable variables—e.g. hairstyle, age, face shape or emotional variables, such as the presence of a smile. All presented labels are the consensus choice among 300 human raters. On average, across all 11 units discovered by the β-VAE, 32.1% of the participants agreed on a single distinct semantic label per unit when presented with a choice of 17 options including "none of the above" (significantly above the 11.82% chance level, $p = 0.00011$; minimum agreement per unit 10.9%, maximum agreement per unit 70.5%; see "Methods"), thus validating the human interpretability of the latents discovered by the β-VAE. These individual β-VAE units were also able to explain the response variance in single recorded neurons, as shown in Fig. 2b. For example, neuron 117 is shown to be sensitive to gender, and neuron 136 is shown to respond differentially to the presence of a smile.

To quantify this effect, we used a metric recently proposed in the machine learning literature, referred to as neural "alignment" in this work for more intuitive exposition, which measures the extent to which variance in each neuron's firing rate can be explained by a single latent unit[29], but is insensitive to the converse, i.e. whether a single unit predicts the response of many biological neurons (Fig. 3a, see "Methods"). The alignment score measures whether the representational form within a subset of the neural population is similar to the representational form

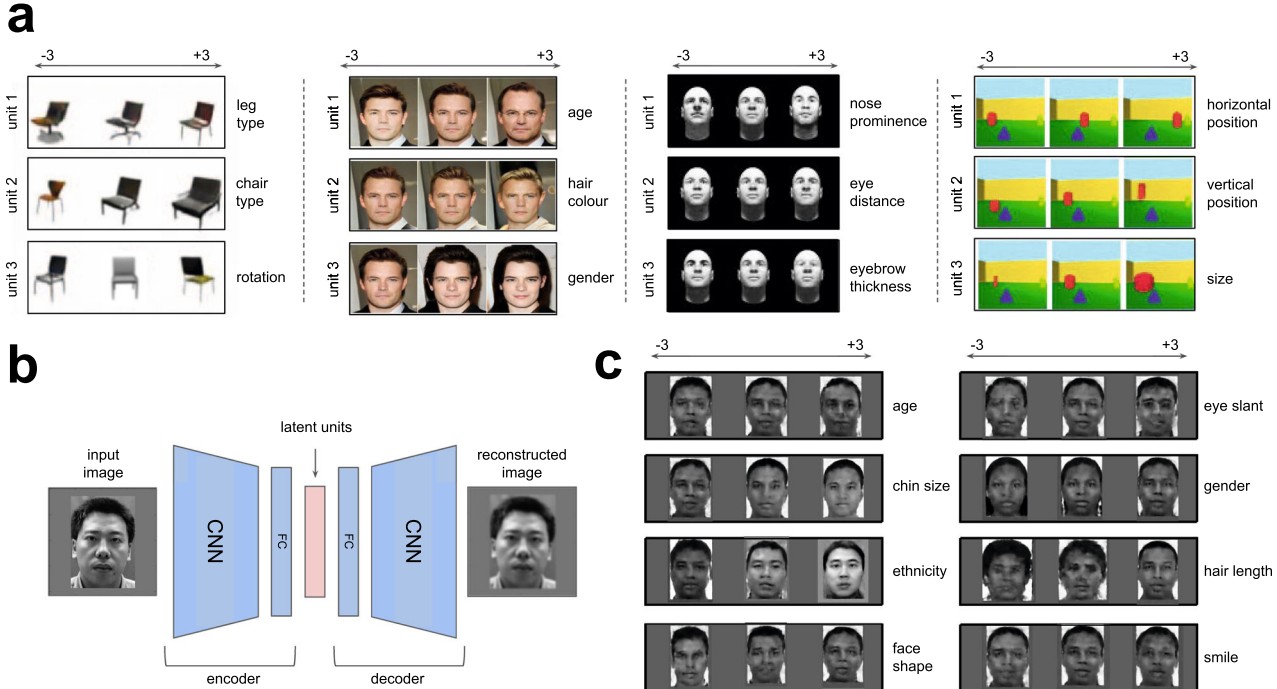

**Fig. 1 Disentangled representation learning. a** Latent traversals used to visualise the semantic meaning encoded by single disentangled latent units of a trained model. In each row the value of a single latent unit is varied between −3 and 3, while the other units are fixed. The resulting effect on the reconstruction is visualised. Each column represents a different model trained to disentangle a different dataset. Chair and face images in the leftmost two traversals are reproduced with the permission of Lee et al.[19]. Traversals of 3D scenes are reproduced with the permission of Burgess et al.[18]. **b** Schematic representation of a self-supervised deep neural network. The encoder maps the input image into a low-dimensional latent representation, which is used by the decoder to reconstruct the original image. Blue indicates trainable neural network units that are free to represent anything. Pink indicates latent representation units that are compared to neurons. CNN, convolutional neural network. FC, fully connected neural network. Face image reproduced with permission from Gao et al.[57]. **c** Latent traversals of eight units of a $\beta$-VAE model trained to disentangle 2100 natural face images. The initial values of all latent units were obtained by encoding the same input image.

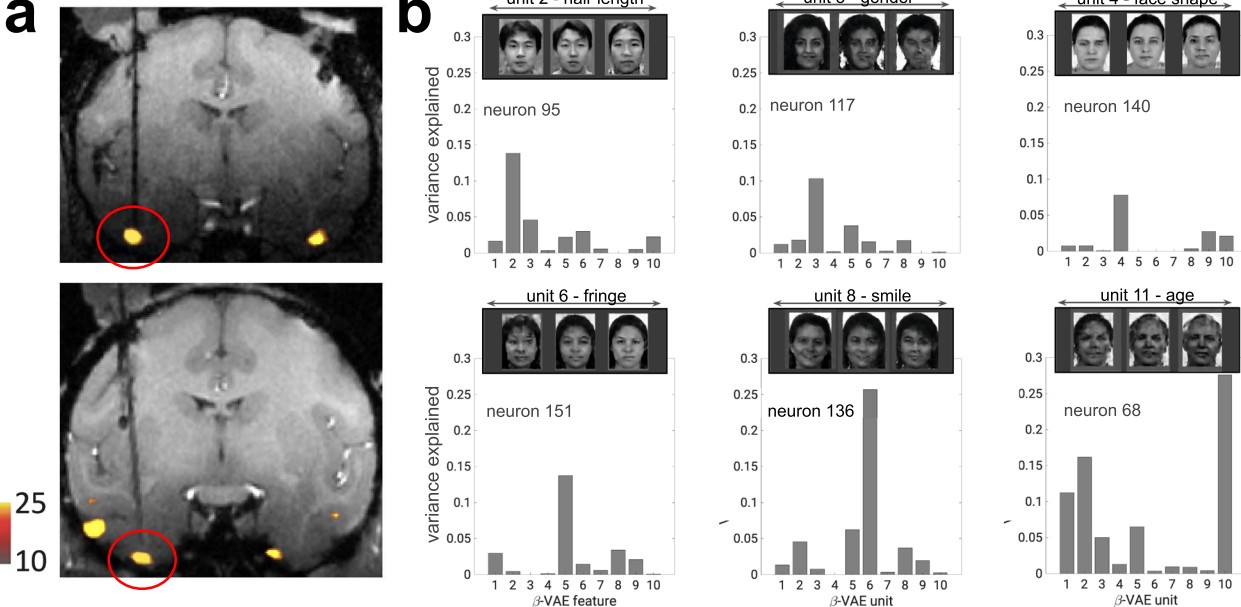

**Fig. 2 Responses of single neurons are well explained by single disentangled latent units. a** Coronal section showing the location of fMRI-identified face patches in two primates, with patch AM circled in red. Dark black lines, electrodes. Reproduced with permission from Chang et al.[6]. **b** Explained variance of single neuron responses to 2100 faces. Response variance in single neurons is explained primarily by single disentangled units encoding different semantically meaningful information (insets, latent traversals as in Fig. 1a, c). Source data are provided as a Source Data file.

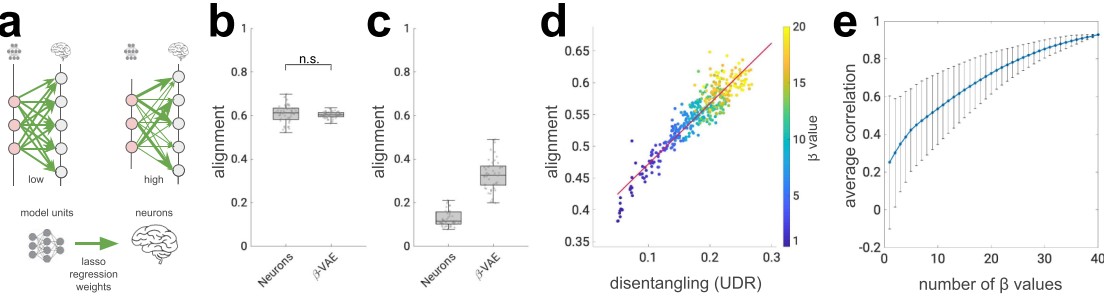

**Fig. 3 Strong alignment between single neurons and disentangled units. a** Schematic of alignment score[29, 63]. Green arrows, lasso regression weights obtained from predicting neural responses from model units (thickness indicates weight magnitude). High alignment scores are obtained when per-neuron regression weights have low entropy (one strong weight); high entropy (all incoming weights are of equal magnitude) results in low alignment scores. **b** $\beta$-VAE alignment scores match the ceiling provided by subsets of neurons ($p = 0.4345$, two-sided Welsch's $t$-test). Circles, alignment per model ($n = 51$) or neuron subsets ($n = 50$). Boxplot centre is median, box extends to 25th and 75th percentiles, whiskers extend to the most extreme data that are not considered outliers, outliers are plotted individually. Source data are provided as a Source Data file. **c** Alignment scores per model ($n = 51$) or neuron subsets ($n = 50$) against artificial neural responses (linear recombination of original neural responses). Boxplot centre is median, box extends to 25th and 75th percentiles, whiskers extend to the most extreme data that are not considered outliers, outliers are plotted individually. Source data are provided as a Source Data file. **d** Alignment scores correlate with the disentanglement quality of latent units obtained from 400 $\beta$-VAE models trained with different $\beta$ values (indicated by colour). UDR, Unsupervised Disentanglement Ranking[31], measures the quality of disentanglement, higher is better. Red line, least squares fit ($r = 0.96$, Pearson correlation). Source data are provided as a Source Data file. **e** Running correlation between UDR and alignment scores across subsets of models. Models in each subset were trained with different $\beta$ values, with the number of $\beta$ values in each subset indicated on the $x$-axis. Rightmost circle, Pearson correlation across 400 $\beta$-VAE models, spanning 40 $\beta$ values as reported in (**d**). Leftmost circle, average across 40 Pearson correlations, each calculated with 10 models with a single $\beta$ value. Bars, standard deviation. Source data are provided as a Source Data file.

discovered by the model, which is a different yet complementary goal to other measures commonly used in the literature[8,28], which instead measure the amount of linearly accessible information shared by the neural and model representations at the population level, while being insensitive to the representational form. High alignment scores indicate that a neural population is intrinsically low-dimensional, with the factors of variation mapping onto the variables discovered by the latent units of the neural network[15].

We first compared alignment scores between the $\beta$-VAE and the monkey data to a theoretical ceiling which was obtained by subsampling the neural data to match the intrinsic dimensionality of the $\beta$-VAE latent representation (see "Methods") and computing its alignment with itself (Fig. 3b). The first surprising observation is that neuron subsets do not reach the maximal alignment score of 1 to the full set of 159 neurons, which points to a significant amount of redundancy in the coding preferences within the neural population. If all neurons encoded unique information, then each neuron in the sampled subset would align uniquely and one-to-one only to itself in the full neural population, resulting in the perfect alignment score. Lower alignment scores indicate that there are a number of other neurons in the population with similar coding properties, resulting in a few-to-one mapping. The second interesting observation is that alignment scores in the $\beta$-VAE met the ceiling provided by the neural subsets, with no reliable difference between the two estimates obtained when the analysis was repeated on multiple subsamples and with multiple network instances ($p = 0.4345$, two-sided Welch's $t$-test). This suggests that each trained $\beta$-VAE instance was able to automatically discover through learning a small number of disentangled units with response properties that are equivalent to the equally sized subsets of real neurons. Furthermore, when we repeated this analysis while computing alignment against fictitious neural responses obtained by linearly recombining the original neural data, we found a significant drop in scores for both the $\beta$-VAE and neural subsets (Fig. 3c, $p = 3.2197e-29$ for $\beta$-VAE, $p = 5.6624e-60$ for neuron subsets; two-sided Welch's $t$-test), indicating that the individual disentangled units discovered by the $\beta$-VAE map significantly better onto the responses of single

neurons recorded from macaque IT, rather than onto their linear combinations. Indeed, the average $\beta$-VAE alignment scores from Fig. 3c are almost as low as those of the random baseline matched in sparsity to the trained $\beta$-VAE instances shown below.

The extent to which the $\beta$-VAE is effective in disentangling a dataset into its latent factors can vary substantially with the way it is regularised, as well as with randomness in its initialisation and training conditions[30]. The parameter after which the network class is named determines the weight of a regularisation term that aims to keep the latent factors independent. Networks with higher values of $\beta$ thus typically give rise to more disentangled representations, as measured by a metric known as the Unsupervised Disentanglement Ranking (UDR, see "Methods")[31], a finding we replicate here. However, we also found that networks with higher UDR scores additionally had higher alignment scores with the neural data (Fig. 3d), and that this relationship held for networks with the same and different values of $\beta$ (Fig. 3e). In other words, the better the network was able to disentangle the latent factors in the face dataset, the more those factors were expressed in single neurons recorded from macaque IT.

**All aspects of the disentanglement objective are important.** Next, we compared the $\beta$-VAE alignment scores with a number of rival models. These baseline models were carefully chosen to disambiguate the role played by the different aspects of the $\beta$-VAE design and training in explaining the coding of neurally aligned variables in its single latent units (see "Methods"). We included a state-of-the-art deep supervised network (VGG)[32] that has previously been proposed as a good model for comparison against neural data in face recognition tasks[33,34], other generative models, such as a basic autoencoder (AE)[35] and a variational autoencoder (VAE)[36], as well as baselines provided by ICA, PCA and a classifier which used only the encoder from the $\beta$-VAE. We defined "latent units" as those emerging in the deepest layers of these networks and, where appropriate, used PCA or feature subsampling (e.g. for VGG raw) to equate the dimensionality of the latent units (to ≤50) to provide a fair comparison with the $\beta$-VAE. We also compared $\beta$-VAE to the "gold standard" provided by the previously published AAM[3], which

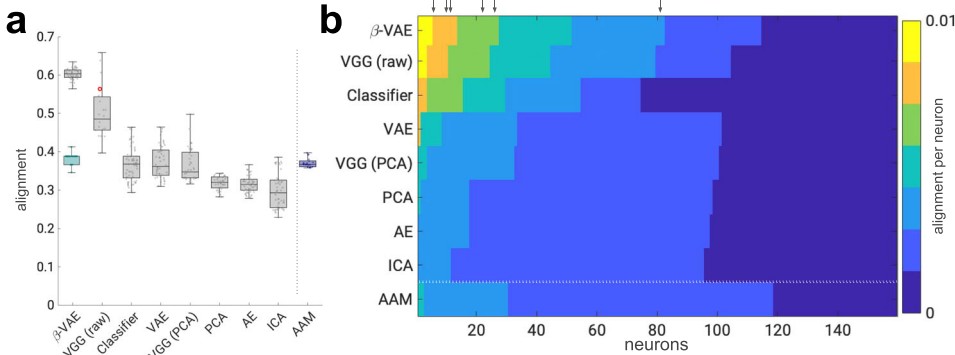

**Fig. 4 Disentangled latent units align with single neurons better than baselines. a** Alignment scores are significantly higher for the β-VAE than the baseline models and the "gold standard" provided by the AAM (all $p < 0.01$, two-sided Welsch's $t$-test; VGG (raw) $p = 7.4220e{-}06$, Classifier $p = 0.0$, AAM $p = 9.1757e{-}42$, VAE $p = 9.7811e{-}43$, VGG (PCA) $p = 1.9383e{-}35$, PCA $p = 0.0$, AE $p = 0.0$, ICA $p = 0.0$). Circles, alignment per model (β-VAE, $n = 51$; VGG (raw), $n = 22$; Classifier, $n = 64$; VAE, Variational AutoEncoder[36], $n = 50$; AE, AutoEncoder[35], $n = 50$; VGG (PCA)[32], $n = 41$; PCA, $n = 41$; ICA, $n = 50$; AAM, active appearance model[3], $n = 21$). Red circle indicates VGG (raw) with all $N = 4096$ units from the last hidden layer. Teal boxplot—random baseline with sparseness matched to the 51 β-VAE models. Boxplot centre is median, box extends to 25th and 75th percentiles, whiskers extend to the most extreme data that are not considered outliers, outliers are plotted individually. Source data are provided as a Source Data file. **b** Per-neuron alignment scores. Scores are discretised into equally spaced bins. Scores in each row are arranged in descending order. The results from single models, chosen to have the median alignment score. VGG (raw) results are presented from the model that contained all $N = 4096$ units from the last hidden layer. Arrows point to neurons from Fig. 2b within the β-VAE alignment scores. Source data are provided as a Source Data file.

produced a low-dimensional code that explained the responses of single neurons to face images well[3,6]. Unlike the β-VAE, which relied on a general learning mechanism to discover its latent units, AAM relied on a manual process idiosyncratic to the face domain. Hence, β-VAE provides a learning-based counterpart to the handcrafted AAM units that could generalise beyond the domain of faces. Although the baselines considered varied in their average alignment scores (Fig. 4a), none approached those of the β-VAE, for which alignment was statistically higher than every other model (all $p$-values $< 0.01$, two-sided Welch's $t$-test). Furthermore, high β-VAE alignment scores could not be explained solely by the sparse nature of disentangled representations, with the random baseline matched in sparseness to the trained β-VAE instances obtaining significantly lower alignment scores (Fig. 4a, teal). The alignment scores broken down by individual neurons are plotted in Fig. 4b for the β-VAE and its baselines.

We validated the findings above using a more direct metric for the coding of latent factors in single neurons, which compared the ratio between the maximum correlation between spike rates and activations in each latent unit, and the sum of such correlations over the model units (average correlation ratio in Fig. 5a, see "Methods"). This ratio was higher for the β-VAE than for other models, confirming the results with alignment scores (Fig. 5b). Interestingly, different neurons did not tend to covary with the same β-VAE latent unit. In fact, there was more heterogeneity among β-VAE units that achieved maximum correlation with the neural responses than among the equivalent units for other models (Fig. 5c). Rich heterogeneity in response properties of single neurons (or latent units) is exactly what would be desired to enable a population of computational units to encode the rich variation in the image dataset. This broad pattern of the results also held when the models were presented with 62 novel face identities that were never seen by the models during training (see Figs. S1b and S2b, c).

Taken together, these results suggest that no one feature of the β-VAE—its training objective (baselined by AE, VAE and classifier), architecture and training data distribution (baselined by VGG) or isolated aspects of its learning objective (baselined by PCA, ICA and the sparse random model)—was sufficient to explain the coding of neurally aligned latent variables in single

units. Rather, it was all of these design choices together that allowed the β-VAE to learn a set of disentangled latent units that explained the responses to single neurons so well.

**Disentanglement discovers a subset of all face dimensions**. So far we have shown that the disentangled representational form in the β-VAE is a closer match to the representational form of real neurons compared to the alternatives presented by the baselines. This, however, is not the whole story. A complementary question to ask is to what degree the information captured by the β-VAE representation overlaps with the information captured by the neural population (see Fig. 6a). Past work demonstrated that at lower dimensionality β-VAE representations can match the "gold standard" AAM model in terms of how well they can explain neural responses at the population level—a result reported in Chang et al.[6]. Here we found that single β-VAE units are able to account for more than 50% of the neuronal variance explained by all 50 units of the highest scoring AAM baseline for around 10% of all neurons (see Fig. 6b and Supplementary Fig. 3). This further corroborates the results in Fig. 3b that suggest that the representations discovered by β-VAE are equivalent to a similarly sized subset of original neurons.

We also calculated how much of the total neural response variance was explained by the models (encoding variance explained), and how much variance in the model responses was explained by the neurons (decoding variance explained) at the population level (see Fig. 6c, d). While the absolute values presented are artificially low due to the lack of noise normalisation (see "Methods"), our results are consistent with those reported in Chang et al.[6]—β-VAE representations contain less information in general than the other baselines (apart from the Classifier and VGG raw), but the information that does get preserved by the β-VAE overlaps with the information within the neural population the most compared to the other baselines (apart from AE). Taken together, the results presented in Fig. 6c, d suggest that in terms of linearly decodable information overlap with the neural population, the best models are AE and β-VAE (closer to the bottom left quadrant in Fig. 6a, while the other baselines are closer to the top left or right quadrants).

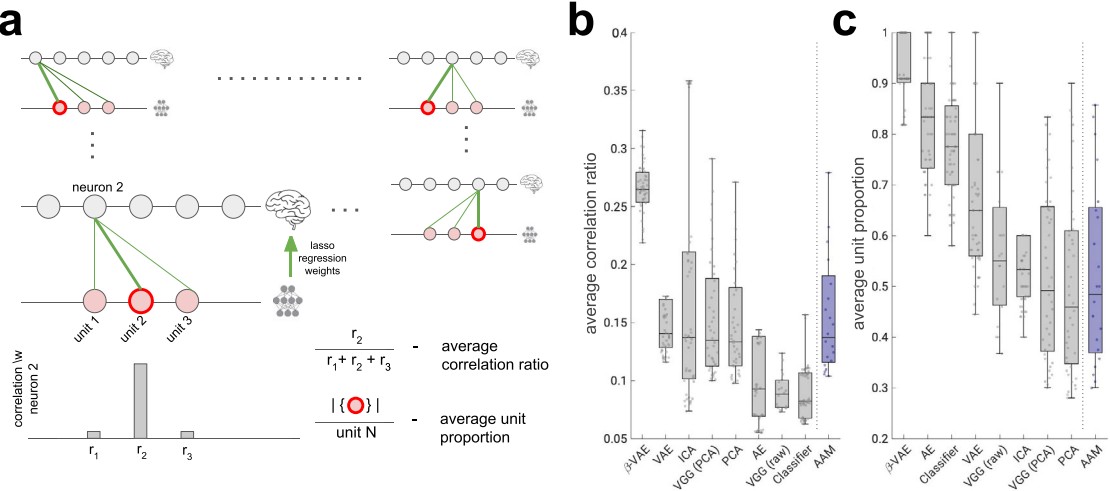

**Fig. 5 Disentangled latent units have better diversity and one-to-one correlation with neurons compared to baselines. a** Schematic of average correlation ratio and average unit proportion scores. A good computational model for explaining responses of single neurons should allow each neuron (grey circle) to be decodable from a single latent unit (pink circle). Green lines are Lasso regression weights as in Fig. 3a. The response of each neuron should correlate strongly with the response of only one latent unit (grey bars) as measured by the average correlation ratio (higher is better). Different neurons should correlate strongly with diverse single latent units (red circles) as measured by the average unit proportion score (higher is better). **b** Average correlation ratio scores are significantly higher for the $\beta$-VAE than the baseline models and the AAM model (all $p < 0.01$; AAM $p = 3.4626e{-}07$, ICA $p = 4.3804e{-}07$, VGG (PCA) $p = 2.9275e{-}15$, PCA $p = 1.8577e{-}15$, VAE $p = 5.7259e{-}13$, AE $p = 8.2339e{-}16$, Classifier $p = 7.8206e{-}55$, VGG (raw) $p = 1.4072e{-}26$, two-sided Welsch's $t$-test). Circles, average correlation ratio score per model ($\beta$-VAE, $n = 51$; VGG (raw), $n = 22$; Classifier, $n = 64$; VAE, Variational AutoEncoder[36], $n = 50$; AE, AutoEncoder[35], $n = 50$; VGG (PCA)[32], $n = 41$; PCA, $n = 41$; ICA, $n = 50$; AAM, active appearance model[3], $n = 21$). Boxplot centre is median, box extends to 25th and 75th percentiles, whiskers extend to the most extreme data that are not considered outliers, outliers are plotted individually. Source data are provided as a Source Data file. **c** Average unit proportion scores are significantly higher for the $\beta$-VAE than the baseline models and the AAM model (all $p < 0.01$; Classifier $p = 4.5163e{-}10$, AE $p = 1.4289e{-}04$, VAE $p = 4.1792e{-}14$, ICA $p = 1.1129e{-}19$, VGG (PCA) $p = 1.7441e{-}19$, PCA $p = 3.5554e{-}19$, AAM $p = 1.8075e{-}09$, VGG (raw) $p = 1.2049e{-}08$, two-sided Welsch's $t$-test). Circles, average unit proportion score per model ($\beta$-VAE, $n = 51$; VGG (raw), $n = 22$; Classifier, $n = 64$; VAE[36], $n = 50$; AE[35], $n = 50$; VGG (PCA)[32], $n = 41$; PCA, $n = 41$; ICA, $n = 50$; AAM[3], $n = 21$). Boxplot centre is median, box extends to 25th and 75th percentiles, whiskers extend to the most extreme data that are not considered outliers, outliers are plotted individually. Source data are provided as a Source Data file.

**Disentangled units are sufficient to decode novel faces.** Finally, we conducted an analysis that sought to link the virtues of the $\beta$-VAE as a tool in machine learning—its capacity to make strong inferences about held out data, with its qualities emphasised here as a theory of visual cognition—strong one-to-one alignment between individual neural and individual disentangled latent units. During training we omitted 62 faces that had been viewed by the monkeys from the training set of the $\beta$-VAE, allowing us to verify that these were reconstructed more faithfully by the $\beta$-VAE than by other networks. Critically, in order to reconstruct these faces, we applied the decoder of the $\beta$-VAE not to its latent units as inferred by its encoder, but rather to the latent unit responses predicted from the activity of a small subset of single neurons (as few as 12) that best aligned with each model unit on a different subset of data (Fig. 7a, see "Methods"). We found that such one-to-one decoding of latent units from the corresponding single neurons was significantly more accurate for the disentangled latent units learnt by the $\beta$-VAE compared to the latent units learnt by other baseline models (all $p$-values $< 0.01$, two-sided Welch's $t$-test) (Fig. 7b). Furthermore, we visualised the $\beta$-VAE reconstructions decoded from just 12 matching neurons (Fig. 7c). Qualitatively, these appeared both more identifiable and of higher image quality than those produced by the latent units decoded from the nearest rival model, the AE, and comparable to those of the basic VAE (as validated by the subjective judgements obtained from 300 human participants, see "Methods"). These results suggest that both the small subset of just 12 neurons and the corresponding 12 disentangled units carried sufficient information to decode previously unseen faces—a stronger result than

past work which required a 1024-dimensional VAE representation to decode novel faces from fMRI voxels[37].

Furthermore, it should be noted that the AE and VAE were explicitly optimised for reconstruction quality resulting in better reconstruction performance than $\beta$-VAE (which was optimised for disentangling at the cost of reconstruction quality[17]) at matched dimensionality during training (all $p < 0.01$, two-sided Welsch's $t$-test, see Supplementary Fig. 1a). Yet, both AE and VAE required more than twice as many neurons as $\beta$-VAE for best decoding from neural data. This, together with the results in ref. 37, suggests that the representational form can matter more than information content for improved decoding performance.

**Discussion**

The results we have presented here validate past evidence[3,6] that the code for facial identity in the primate IT is low-dimensional, with single neurons encoding independent axes of variation. Unlike the previous work[3,6], however, our results demonstrate that such a code can also be semantically interpretable at a single neuron level. In particular, we show that the axes of variation represented by single IT neurons align with single "disentangled" latent units that appear to be largely semantically meaningful and which are discovered by the $\beta$-VAE, a recent class of deep neural networks proposed in the machine learning community that does not rely on extensive teaching signals for learning. Given the strong alignment of single IT neurons with the single units discovered through disentangled representation learning, and the fact that disentangling can be done through self-supervision without the need for an external teaching

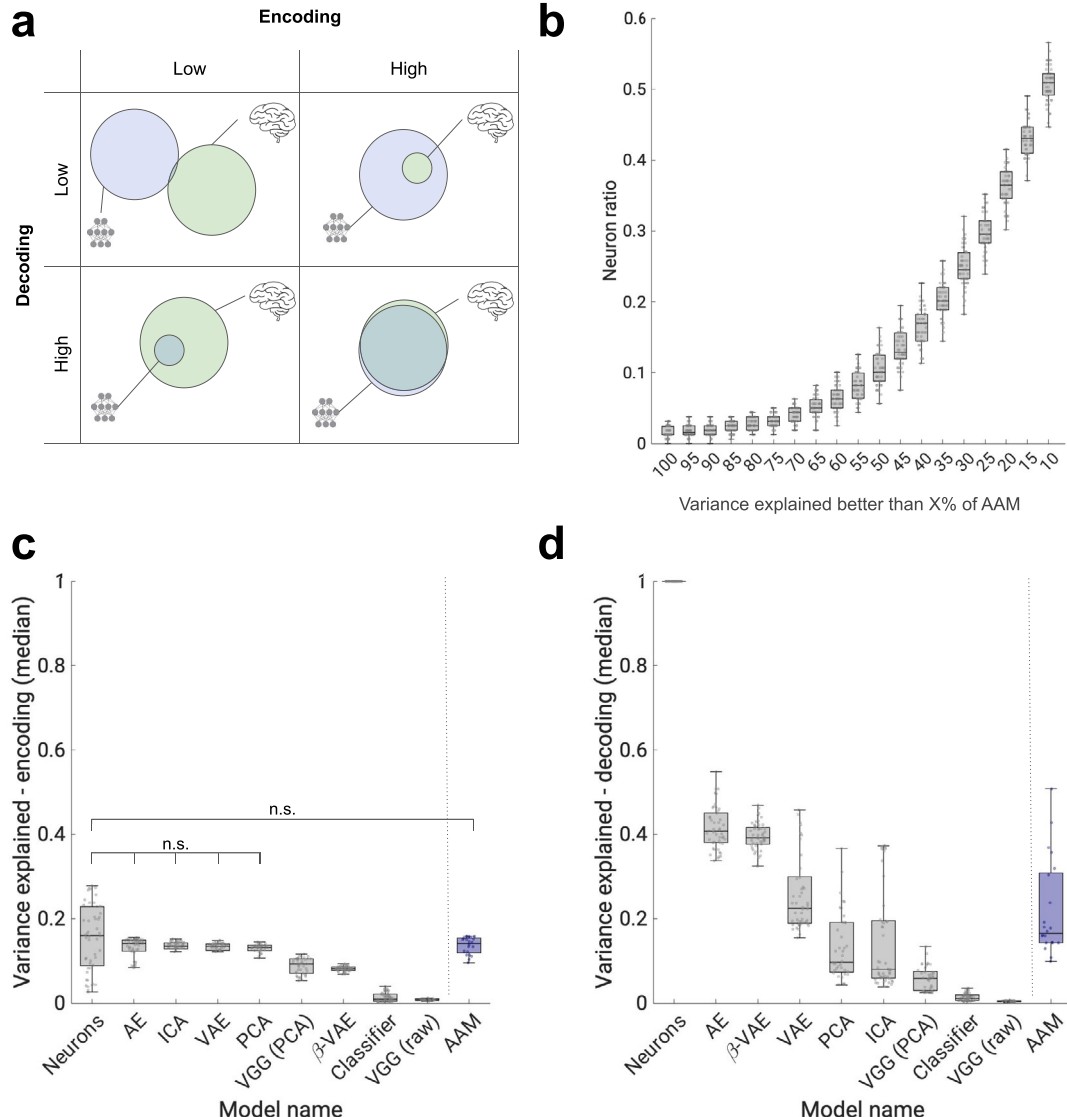

**Fig. 6 Variance-explained results. a** Schematic representation of linearly accessible information overlap between the population of neurons (green) and model representation (blue) corresponding to the different combinations of magnitudes of the encoding and decoding scores. **b** Ratio of total neuron population ($n = 159$) for which single $\beta$-VAE units explain more variance than $X\%$ of variance explained by the best baseline model (AAM, 50 units). See Supplementary Fig. 3 for more details. Source data are provided as a Source Data file. **c** Encoding variance explained. No significant difference is found between encoding variance explained by neuron subsets and AE, ICA, VAE, PCA and AAM ($\beta$-VAE $p = 2.4689e{-}08$, AAM $p = 0.1211$, VGG (PCA) $p = 2.5055e{-}07$, PCA $p = 0.0174$, Classifier $p = 2.6311e{-}17$, ICA $p = 0.0185$, AE $p = 0.0368$, VAE $p = 0.0176$, VGG (raw) $p = 8.4568e{-}18$, two-sided Welsch's $t$-test). Circles, median explained variance across 159 neurons ($\beta$-VAE, $n = 51$; VGG (raw), $n = 22$; Classifier, $n = 64$; VAE, Variational AutoEncoder[36], $n = 50$; AE, AutoEncoder[35], $n = 50$; VGG (PCA)[32], $n = 41$; PCA, $n = 41$; ICA, $n = 50$; AAM, active appearance model[3], $n = 21$). Boxplot centre is median, box extends to 25th and 75th percentiles, whiskers extend to the most extreme data that are not considered outliers, outliers are plotted individually. Source data are provided as a Source Data file. **d** Decoding variance explained. $\beta$-VAE variance explained is statistically significantly different from all other models (all $p < 0.01$; AAM $p = 7.6510e{-}07$, VGG (PCA) $p = 0.0$, PCA $p = 6.0566e{-}23$, Classifier $p = 0.0$, ICA $p = 2.8584e{-}20$, AE $p = 0.0164$, VAE $p = 1.1390e{-}12$, VGG (raw) $p = 0.0$, two-sided Welsch's $t$-test). Circles, median explained variance across model units ($\beta$-VAE, $n = 51$; VGG (raw), $n = 22$; Classifier, $n = 64$; VAE, Variational AutoEncoder[36], $n = 50$; AE, AutoEncoder[35], $n = 50$; VGG (PCA)[32], $n = 41$; PCA, $n = 41$; ICA, $n = 50$; AAM, active appearance model[3], $n = 21$). Boxplot centre is median, box extends to 25th and 75th percentiles, whiskers extend to the most extreme data that are not considered outliers, outliers are plotted individually. Source data are provided as a Source Data file.

signal, it is plausible that the ventral visual stream may also be optimising the disentangling learning objective.

Our work extends recent studies of the coding properties of single neurons in the primate face-patch area, reporting finding one-to-one correspondences between model units and neurons, as opposed to few-to-one as previously reported[3]. Moreover, we show that disentangling may occur at the end of the ventral visual stream (IT), extending the results recently reported for V1[38]. Past studies have proposed that the ventral visual cortex may

disentangle[1,38] and represent visual information with a low-dimensional code[3,39,40]. However, this work did not ask how these representations emerge via learning. Here, we propose a theoretically grounded[41] computational model (the $\beta$-VAE) for how disentangled, low-dimensional codes may be learnt from the statistics of visual inputs[17].

It is worth noting that, in general, the baselines, including AAM, contain a larger number informative dimensions than $\beta$-VAE. Furthermore, it has been demonstrated that when the larger

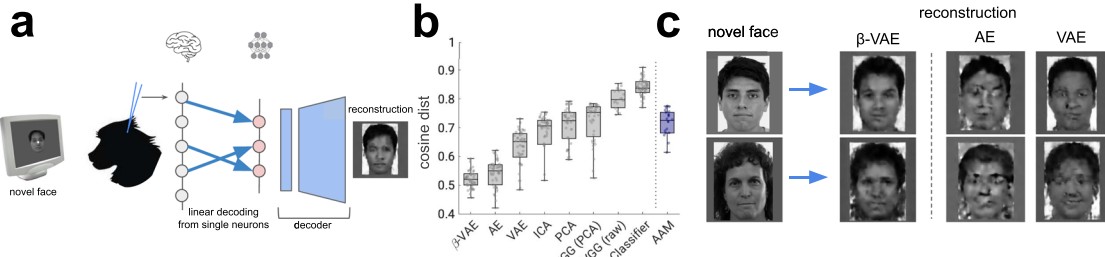

**Fig. 7 Reconstructing novel faces from single neurons. a** Responses of 159 neurons (grey circles) in face-patch area AM were recorded while two primates viewed 62 novel faces. One-to-one match was found between each model unit (pink circles) and a corresponding single neuron. Linear regression (blue arrow) was used to decode the responses of each individual model latent unit (pink circles) from the activations of its corresponding single neuron. The pre-trained model decoder was used to reconstruct the novel face. Face image reproduced with permission from Chang et al.[6]. **b** Cosine distance between real standardised latent unit responses and those decoded from single neurons are significantly smaller for $\beta$-VAE compared to baseline models and the "gold standard" provided by the AAM model (all $p < 0.05$, single-sided Welsch's $t$-test; AE $p = 0.0195$, VAE $p = 1.0596e{-}14$, PCA $p = 4.0370e{-}25$, ICA $p = 1.5758e{-}12$, VGG (PCA) $p = 5.0467e{-}24$, Classifier $p = 0.0$, AAM $p = 2.3216e{-}14$, VGG (raw) $p = 4.8840e{-}20$). Circles, median cosine distance per model ($\beta$-VAE, $n = 51$; VGG (raw), $n = 22$; Classifier, $n = 64$; VAE, Variational AutoEncoder[36], $n = 50$; AE, AutoEncoder[35], $n = 50$; VGG (PCA)[32], $n = 41$; PCA, $n = 41$; ICA, $n = 50$; AAM, active appearance model[3], $n = 21$). Boxplot centre is median, box extends to 25th and 75th percentiles, whiskers extend to the most extreme data that are not considered outliers, outliers are plotted individually. Source data are provided as a Source Data file. **c** $\beta$-VAE can decode and reconstruct novel faces from 12 matching single neurons. The reconstructions are better than those from the closest baselines, AE and VAE, which required 30 and 27 neurons for decoding, respectively. The $\beta$-VAE instance was chosen to have the best disentanglement quality as measured by the UDR score; AE and VAE instances were chosen to have the highest reconstruction accuracy on the training dataset. Face images reproduced with permission from Ma et al.[53] and Phillips et al.[55].

full set of AAM dimensions is used, it explains more neural variance at the population level than the smaller full set of disentangled $\beta$-VAE units[6]. Indeed, it is clear that the handful of disentangled dimensions discovered by $\beta$-VAE in the current study are not sufficient to fully describe the whole space of faces. The reason why $\beta$-VAE discovers only a subset of the dimensions necessary to describe faces fully stems from a known limitation of the current methods for disentangled representation learning—they rely on training on well-aligned large datasets for achieving maximal interpretability and disentangling quality. Indeed, applying $\beta$-VAE to a more suitable dataset of faces[42] allows it to recover at least double the number of disentangled dimensions than the number found in the current study (see Supplementary Information for more details, example latent traversals shown in Fig. 1a, second from the right). The lack of appropriate levels of scale and alignment for the dataset of 2100 faces used in this study leaves room for improvements both in terms of latent interpretability and the amount of population neural variance explained to future work.

Saying this, an important aspect of our proposed learning mechanism is that it generalises beyond the domain of faces[17–19]. We believe that the difficulty in identifying interpretable codes in the IT encountered in the past may have been due to the fact that semantically meaningful axes of variation of complex visual objects are more challenging for humans to define (and hence use as visual probes) compared to simple features, such as visual edges[14]. A computational model like the $\beta$-VAE, on the other hand, is able to automatically discover disentangled latent units that align with such axes, as was demonstrated for the domain of faces in this work. Hence, we hope that the neuroscience community will be able to take advantage of any further advancements in disentangled representation learning techniques within the machine learning community to study neural responses in the IT beyond the domain of faces. Using face perception as the test domain for building the connection between neural coding in the IT and disentangling deep generative models as was done in this work has unique advantages. Specifically, both neural responses and image statistics in this domain have been particularly well studied compared to other visual stimulus classes. This allows for comparisons with strong hand-engineered baselines[3] using

relatively densely sampled neural data[5]. Furthermore, although faces make up a small subset of all possible visual objects, and neurons that preferentially respond to faces tend to cluster in particular patches of the IT cortex[5], the computational mechanisms and basic units of representation employed for face processing may in fact generalise more broadly within the ventral visual stream[5,43]. Indeed, face perception is seen by many to be a "microcosm of object recognition"[5]. Hence, assuming that this is the case, $\beta$-VAE and any future more advanced methods for disentangled representation learning may serve as a promising tools to understand IT codes at a single neuron level even for rich and complex visual stimuli in the future.

One contribution of this paper is the introduction of novel measures for comparing neural and model representations. Unlike other often used representation comparison methods (e.g. explained variance of neuron-level regressions[8,11,25–27] or representational similarity analysis (RSA)[11,28]) which are insensitive to invertible linear transformations, our methods measure the alignment between individual neurons and model units. Hence, they do not abstract away the representational form and preserve the ability to discriminate between alternative computational models that may otherwise score similarly[15]. To summarise, while the traditional methods compare the informativeness of representations, our approach compares their representational form, hence the two are complementary to each other.

While the development of $\beta$-VAE for learning disentangled representations was originally guided by high-level neuroscience principles[44–46], subsequent work in demonstrating the utility of such representations for intelligent behaviour was primarily done in the machine learning community[22–24,47]. In line with the rich history of mutually beneficial interactions between neuroscience and machine learning[48], we hope that the latest insights from machine learning may now feed back to the neuroscience community to investigate the merit of disentangled representations for supporting intelligence in the biological systems, in particular as the basis for abstract reasoning[49], or generalisable and efficient task learning[50].

## Methods
**Dataset.** We used a dataset of 2162 natural grayscaled, centred and cropped images of frontal views of faces without nuisance obstructions (e.g. facial hair or a

head garment), under normal lighting conditions and without strong facial expressions pasted on a grey $200 \times 200$ pixel background as described in[6]. The face images were collated from multiple publicly available datasets: 72 images from AR Face Database[51], 48 images from CelebA[52], 457 images from Chicago Face Database[53], 64 faces from CVL[54], 563 images from FERET[55], 45 images from MR2[56] and 913 images from PEAL[57] dataset. In all, 62 held out face images were randomly chosen. These faces were among the 2100 faces presented to the primates, but not among the 2100 faces used to train the models. All models (apart from VGG) were trained on the same set of faces, which were mirror flipped with respect to the images presented to the primates. This ensured that the train and test data distributions were similar, but not identical (see Supplementary Fig. 1a). To train the Classifier baseline, we augmented the data with $5 \times 5$ pixel translations of each face to ensure that multiple instances were present for each unique face identity. The data were split into 80%/10%/10% train/validation/test sets.

**Neurophysiological data**. All neurophysiological data were re-used from Chang et al.[6]. The data were collected from two male rhesus macaques (*Macaca mulatta*) of 7–10 years old. The animals were pair-housed and kept on a 14h/10h light/dark cycle. All procedures conformed to local and US National Institutes of Health guidelines, including the US National Institutes of Health Guide for Care and Use of Laboratory Animals. All experiments were performed with the approval of the Caltech Institutional Animal Care and Use Committee (IACUC).

Face patches were identified by finding regions responding significantly more to 16 faces than to 80 non-face stimuli (bodies, fruits, gadgets, hands, and scrambled patterns[4]) while passively viewing images on a screen in a 3T TIM (Siemens, Munich, Germany) magnet. Feraheme contrast agent was injected to improve signal/noise ratio. The results were confirmed across multiple independent scan sessions.

For single-unit recording, tungsten electrodes (18–20 Mohm at 1 kHz, FHC) backloaded into plastic guide tubes set to reach approximately 3–5 mm below the dura surface were used. The electrode was advanced slowly with a manual advancer (Narishige Scientific Instrument, Tokyo, Japan). Extracellular action potentials were isolated using the box method using an online spike sorting system (Plexon, Dallas, TX, USA) from amplified neural signals. Spikes were sampled at 40 kHz. All spike data were further re-sorted offline using Plexon spike sorting clustering algorithm. Only well-isolated units were considered for further analysis. The image stimuli were presented on a CRT monitor (DELL P1130). The intensity of the screen was measured using a colorimeter (PR650, Photo Research) and linearised for visual stimulation. Screen size covered $27.7 \times 36.9$ visual degrees and stimulus size spanned 5.7°. The fixation spot size was 0.2° in diameter and the fixation window was a square with the diameter of 2.5°. The monkeys were head fixed and passively viewed the screen in a dark room. Eye position was monitored using an infrared eye tracking system (ISCAN). Juice reward was delivered every 2–4 s if fixation was properly maintained. Images were presented in random order. All images were presented for 150 ms interleaved by 180 ms of a grey screen. Each image was presented 3–5 times. The number of spikes in a time window of 50–350 ms after stimulus onset was counted for each stimulus and used to calculate the face-selectivity index of each cell according to

$$\mathrm{FSI} = \frac{\overline{\mathbf{n}}_{\mathrm{face}} - \overline{\mathbf{n}}_{\mathrm{non-face}}}{\overline{\mathbf{n}}_{\mathrm{face}} + \overline{\mathbf{n}}_{\mathrm{non-face}}} \qquad (1)$$

where $\overline{\mathbf{n}}$ is the mean activity of a single neuron in response to either face or non-face stimuli. Only neurons with high face selectivity (FSI > 0.33) were selected for further analysis.

**Artificial neurophysiological data**. In order to investigate whether the responses of $\beta$-VAE units encoded linear combinations of neural responses, we created artificial neural data by linearly recombining the responses of the real neurons. We first standardised the responses of the 159 recorded neurons across the 2100 face images. We then multiplied the original matrix of neural responses with a random projection matrix $A$. Each value $A_{ij}$ of the projection matrix was sampled from the unit Gaussian distribution. The absolute value of the matrix was then taken, and each column was normalised to sum to 1.

**Neuron subsets**. For fairer comparison with the models, which learnt latent representations of size $N \in [10, 50]$ as will be described below, we sampled neural subsets with 50 or fewer neurons. To do this, we first uniformly sampled five values from $N \in [10, 50]$ without replacement to indicate the size of the subsets. Then, for each size value we sampled 10 random neuron subsets without replacement, resulting in 50 neuron subsets in total.

**Human participants**. In order to validate the semantic meaningfulness of $\beta$-VAE latent units shown in Figs. 1c and 2b, and the judged quality of model reconstructions shown in Fig. 7c, we recruited 600 human participants (300 for each of the two studies, age $30.81 \pm 11.07$ years, 117 females for identifying transformations applied to faces, and age $30.75 \pm 10.57$ years, 123 females for comparing face reconstructions). The participants were recruited through the Prolific crowd-sourcing platform. The full details of our study design, including compensation rates, were reviewed and approved by DeepMind's independent ethical review

committee. All participants provided informed consent prior to completing tasks and were reimbursed for their time.

**$\beta$-VAE model**. We used the standard architecture and optimisation parameters introduced in[17] for training the $\beta$-VAE (Fig. 8a). The encoder consisted of four convolutional layers ($32 \times 4 \times 4$ stride 2, $32 \times 4 \times 4$ stride 2, $64 \times 4 \times 4$ stride 2 and $64 \times 4 \times 4$ stride 2), followed by a 256-d fully connected layer and a 50-d latent representation. The decoder architecture was the reverse of the encoder. We used ReLU activations throughout. The decoder parameterised a Bernoulli distribution. The model was implemented using TensorFlow 1.0 (e.g. see https://github.com/google-research/disentanglement_lib). We used Adam optimiser with $1e-4$ learning rate and trained the models for 1 mln iterations using batch size of 16, which was enough to achieve convergence. The models were trained to optimise the following disentangling objective:

$$\mathcal{L}_{\beta-\mathrm{VAE}} = \mathbb{E}_{p(\mathbf{x})}[\,\mathbb{E}_{q_{\phi}(\mathbf{z}|\mathbf{x})}[\log p_{\theta}(\mathbf{x}|\mathbf{z})] - \beta KL(q_{\phi}(\mathbf{z}|\mathbf{x})\,||\,p(\mathbf{z}))\,] \qquad (2)$$

where $p(\mathbf{x})$ is the probability of the image data, $q(\mathbf{z}|\mathbf{x})$ is the learnt posterior over the latent units given the data, and $p(\mathbf{z})$ is the unit Gaussian prior with a diagonal covariance matrix. Note that due to the limited amount of training data (2100 images compared to typical dataset sizes approaching 1 mln images), we were not able to achieve the best disentangling or reconstruction performance that this model class if capable of in principle, resulting in fewer disentangled dimensions discovered by each trained model, and choppier reconstruction quality. Saying this, the reported models typically still converged on approximately the same disentangled representation. All the results are reported using disentangled dimensions discovered by single models—we never mix or combine disentangled dimensions across models.

**Baseline models**. We compared $\beta$-VAE to a number of baselines to test whether any individual aspects of $\beta$-VAE training could account for the quality of its learnt latent units. To disambiguate the role of the learning objective, we compared $\beta$-VAE to a traditional autoencoder (AE)[35] and a basic variational autoencoder (VAE)[36,58]. These models had the same architecture, training data, and optimisation parameters as the $\beta$-VAE (Fig. 8a), and were also implemented in TensorFlow 1.0, but their learning objectives were different. The AE optimised the following objective that tried to optimise the quality of its reconstructions:

$$\mathcal{L}_{\mathrm{AE}} = \mathbb{E}_{p(\mathbf{x})}\,||f(\mathbf{x};\theta,\phi) - \mathbf{x}||^2 \qquad (3)$$

where $f(\mathbf{x};\theta,\phi)$ is the image reconstruction produced by putting the original image through the encoder and decoder networks parameterised by $\phi$ and $\theta$, respectively. The VAE optimised the variational lower bound on the data distribution $p(\mathbf{x})$:

$$\mathcal{L}_{\mathrm{VAE}} = \mathbb{E}_{p(\mathbf{x})}[\,\mathbb{E}_{q_{\phi}(\mathbf{z}|\mathbf{x})}[\log p_{\theta}(\mathbf{x}|\mathbf{z})] - KL(q_{\phi}(\mathbf{z}|\mathbf{x})\,||\,p(\mathbf{z}))\,] \qquad (4)$$

where $q(\mathbf{z}|\mathbf{x})$ is the learnt posterior over the latent units given the data, and $p(\mathbf{z})$ is the isotropic unit Gaussian prior.

To test whether the supervised classification objective could be a good alternative to the self-supervised disentangling objective, we compared $\beta$-VAE to two classifier neural network baselines. One of these baselines, referred to as the Classifier in all the figures and the text, shared the encoder architecture, the data distribution and the optimisation parameters with the $\beta$-VAE (Fig. 8b), but instead of disentangling, it was trained to differentiate between the 2100 faces using a supervised objective. In particular, the four convolutional layers and the fully connected layer of the encoder fed into an $N$-dimensional representation, which was followed by 2100 logits that were trained to recognise the unique 2100 face identities. In order to avoid overfitting, we used early stopping. The final models were trained for between 300k and 1mln training steps. This model was also implemented in TensorFlow 1.0.

The other classifier baseline was the VGG-Face model[32] (referred to as the VGG in all the figures and the text), a more powerful deep network developed for state-of-the-art face recognition performance and previously chosen as an appropriate computational model for comparison against neural data in face recognition tasks[33,34,59] (Fig. 8c). Similarly to other works[6,33,34,59], we used a standard pre-trained MatLab implementation (http://www.vlfeat.org/matconvnet/pretrained/) of the VGG network, trained to differentiate between 2622 unique individuals using a dataset of 982,803 images[32]. Note that the data used for VGG training were unrelated to the 2100 face images presented to the primates. The VGG therefore had a different architecture, training data distribution and optimisation parameters compared to the $\beta$-VAE. The model consisted of 16 convolutional layers, followed by 3 fully connected layers (see[32] for more details). The last hidden layer before the classification logits contained 4096 units. Following the precedent set by refs. [6] and[59], we used PCA to reduce the dimensionality of the VGG representation by projecting the activations in its last hidden layer in response to the 2100 test faces to the top $N$ principal components (PCs) (Fig. 8c, referred to as VGG (PCA) in figures). Alternatively, we also randomly subsampled the units in the last hidden layer of VGG (without replacement) to control for any potential linear mixing of their responses which PCA could plausibly introduce (Fig. 8c, referred to as VGG (raw) in figures).

To rule out that the responses of single neurons could be modelled by simply explaining the variance in the data, we compared $\beta$-VAE to $N$ PCs produced by

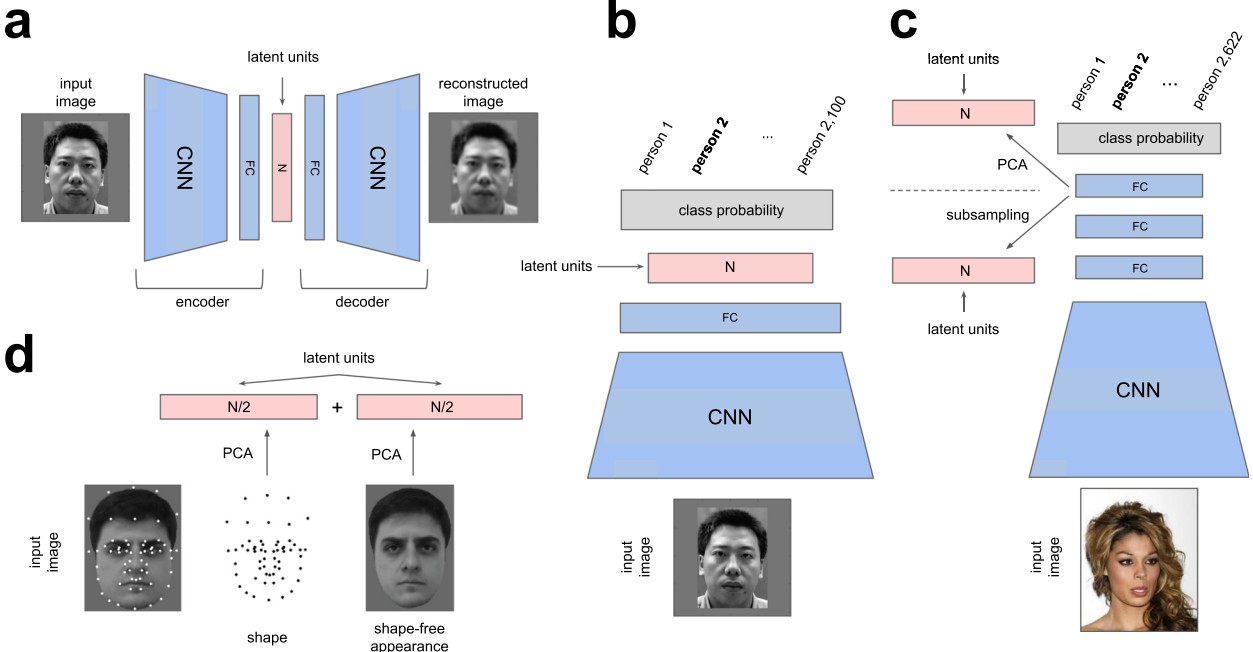

**Fig. 8 Schematic of model architectures.** Blue, trainable neural network units free to represent anything. Pink, latent representation units used for comparison with neurons in response to 2100 face images. Grey, units representing class probabilities. CNN, convolutional neural network. FC, fully connected neural network. N, number of latent units. **a** Self-supervised models — $\beta$-VAE[17], autoencoder (AE)[35] and variational autoencoder (VAE)[36, 58]. Models were trained on the mirror flipped versions of the 2100 faces presented to the primates. Face image reproduced with permission from Gao et al.[57]. **b** Classifier baseline. Encoder network, same as in (**a**). Model trained to differentiate between unique 2100 face identities using mirror flipped versions of the 2100 faces augmented with 5 × 5 pixel translations. Face image reproduced with permission from Gao et al.[57]. **c** VGG baseline[32]. Encoder network has larger and deeper CNN and FC modules than in (**a**) and (**b**). Representation dimensionality is reduced to match other models either by a projection on the first N principal components (PCs) (VGG (PCA)), or by taking a random subset of N units without replacement (VGG (raw)). VGG was trained to differentiate between 2622 unique faces using a face dataset[32] unrelated to the 2100 faces presented to the primates. Face image is representative of the images used to train the model and is reproduced with permission from Liu et al.[52]. **d** Active appearance model (AAM)[3]. Keypoints were manually placed on the 2100 face images. First N/2 PCs over the keypoint locations formed the "shape" latent units. First N/2 PCs over the shape-normalised images formed the "appearance" latent units. Figure adapted with permission from Chang et al.[3].

applying principal component analysis (PCA) to the 2100 faces using sklearn 0.2 PCA package. To rule out the role of simply finding the independent components of the data during $\beta$-VAE training, we compared $\beta$-VAE to the N independent components discovered by independent component analysis (ICA) applied to the 2100 face images using sklearn 0.2 FastICA package.

Finally, we also compared $\beta$-VAE to the active appearance model (AAM). Linear combinations of small numbers of its latent units (six on average) was previously reported to explain the responses of single neurons in the primate AM area well[3,6]. We re-used the AAM latent units from[6]. These were obtained by setting 80 landmarks on each of the 2100 facial images presented to the primates. The positions of landmarks were normalised to calculate the average shape template. Each face was warped to the average shape using spline interpolation. The warped image was normalised and reshaped to a 1-d vector. PCA was carried out on landmark positions and shape-free intensity independently. The first N/2 shape PCs and the first N/2 appearance PCs were concatenated to produce the N-dimensional AAM representations (Fig. 8d).

**Hyperparameter sweep**. To ensure that all models had a fair chance of learning a useful representation, we trained multiple instances of each model class using different hyperparameter settings. The choice of hyperparameters and their values were dependent on the model class. However, all models went through the same model selection pipeline: (1) K model instances with different hyperparameter settings were obtained as appropriate; (2) S ⊆ K models with the best performance on the training objective were selected; (3) models that did not discover any latent units that shared information with the neural responses were excluded, resulting in M ⊆ S models retained for the final analyses. These steps are expanded below for each model class.

For the $\beta$-VAE model, the main hyperparameter of interest that affects the quality of the learnt latent units is the value of $\beta$. The $\beta$ hyperparameter controls the degree of disentangling achieved during training, as well as the intrinsic dimensionality of the learnt latent representation[17]. Typically, a $\beta > 1$ is necessary to achieve good disentangling, however, the exact value differs for different datasets. Hence, we trained 400 models with different values of $\beta$ by uniformly sampling 40 values of $\beta$ in the [0.5, 20] range. Another factor that

affects the quality of disentangled representation is the random initialisation seed for training the models[30]. Hence, for each $\beta$ value, we trained 10 models from different random initialisation seeds, resulting in the total pool of 400 trained $\beta$-VAE model instances to choose from. All $\beta$-VAE models were initialised to have N = 50 latent units, however, due to the variability in the values of $\beta$, the intrinsic dimensionality of the trained models varied between 10 and 50.

In order to isolate the role of disentangling within the $\beta$-VAE optimisation objective from the self-supervision aspect of training, we kept as many choices as possible unchanged between the $\beta$-VAE and the AE/VAE baselines: the model architecture, optimiser, learning rate, batch size and number of training steps. The remaining free hyperparameters that could affect the quality of the AE/VAE learnt latent units were the random initialisation seeds, and the number of latent units N. The latter was necessary to sweep over explicitly, since AE and VAE models do not have an equivalent to the $\beta$ hyperparameter that affects the intrinsic dimensionality of the learnt representation. Hence, we trained 100 model instances for each of the AE and VAE model classes, with five values of N sampled uniformly without replacement from N ∈ [10, 50], each trained from 20 random initialisation seed values.

For the Classifier baseline, we used the following hyperparameters for the initial selection: five values of N ∈ [10, 50] sampled uniformly without replacement, as well as a number of learning rate values {$1e{-}3$, $1e{-}4$, $1e{-}5$, $1e{-}6$, $1e{-}7$} and batch sizes {16, 64, 128, 256}, resulting in 100 instances. We trained the models with early stopping to avoid overfitting, and used the classification performance on the validation set to choose the settings for the learning rate and batch size. We found that the values used for training $\beta$-VAE, AE and VAE (learning rate $1e{-}4$, batch size 16) were also reasonable for training the Classifier, achieving >95% classification accuracy. Hence, we trained the final set of 450 Classifier model instances with fixed learning rate and batch size, five values of N ∈ [10, 50] sampled uniformly without replacement and 50 random seeds.

We used sklearn 0.2 FastICA algorithm[60] to extract ICA units, which is dependent on the random initialisation seed. Hence, we extracted N ∈ [10, 50] independent components with 10 random initialisation seeds each, resulting in 41 ICA model instances.

The remaining baseline models relied on using a single canonical model instance (VGG and AAM) and/or on a deterministic dimensionality reduction process (PCA, AAM, VGG). Hence, the random seed hyperparameter did not apply to them. In order to make a fairer comparison with the other baselines, we therefore created different model instances by extracting different numbers of representation dimensions with $N \in [10, 50]$, resulting in 41 PCA and VGG (PCA) model instances, and 21 AAM instances (since $N$ needs to split evenly into shape- and appearance-related units). For the VGG (raw) variant, we first uniformly sampled five values from $N \in [10, 50]$ without replacement to indicate the size of the hidden unit subsets. Then, for each size value we sampled 10 random hidden unit subsets without replacement, resulting in 50 VGG (raw) model instances in total.

**Model selection based on training performance.** For each model class, apart from the deterministic baselines (PCA, AAM and VGG), we selected a subset of model instances based on their training performance. For the $\beta$-VAEs, we used the recently proposed Unsupervised Disentanglement Ranking (UDR) score[31] implemented in MatLab R2017b to select 51 model instances with the most disentangled representations (within the top 15% of UDR scores) for further analysis. For AE baseline, we selected 50 model instances with the lowest reconstruction error per chosen value of $N$. For the VAE baseline we selected 50 model instances with the highest lower bound on the training data distribution per chosen value of $N$. Finally, for the Classifier baseline, we selected 81 models which achieved >95% classification accuracy on the test set.

**Filtering out uninformative models.** To ensure that all models used in the final analyses shared at least some information with the recorded neural population, we performed the following filtering procedure. First, we trained Lasso regressors (MatLab R2017b) as per Variance Explained section below, to predict the responses of each neuron across the 2100 faces from the population of latent units extracted from each trained model. We then calculated the mean amount of variance explained (VE) averaged across all neurons for each of the models. We filtered out all models where $VE < \overline{VE} - SD(VE)$, with $\overline{VE}$ and $SD(VE)$ represent the mean and standard deviation of VE scores across all models, respectively.

The full-model selection pipeline resulted in 51 $\beta$-VAE model instances, 50 AE, VAE and ICA model instances, 41 PCA and VGG (PCA) model instances, 22 VGG (raw) model instances, 21 AAM model instances and 64 Classifier model instances that were used for further analyses.

**Human psychophysics: identifying transformations applied to faces.** While there is no readily available metric via which we can quantify the interpretability of $\beta$-VAE latent units, we designed a psychophysical study to test the null hypothesis that people cannot agree upon the variable coded by each latent unit. To validate the semantic meaningfulness of $\beta$-VAE latent units we asked 300 participants to select which of the 17 provided options best described the transformation generated by traversing a single latent unit. The following label options were presented to the participants: age, chin size, ethnicity, eye distance, eye size, eye slant, eyebrow position, face length, face shape, forehead size, fringe/bangs, gender, hair density, hair length, nose size, smiling and none of the above. Since all well-disentangled $\beta$-VAE models learnt approximately the same representation, we found corresponding units in two trained and well-disentangled $\beta$-VAE models and presented traversals from the two corresponding units applied to two different faces for the participants to label. The participants were presented with the following instructions: "You will be asked to identify 11 transformations applied to faces. For each transformation, you will be provided with two examples containing two different faces, each one transformed in the same way. You will be asked to make a judgement of what the transformation is, and select an answer from the drop-down list that best matches your guess.". Each face transformation was generated by traversing the value of one chosen unit of a pre-trained $\beta$-VAE between $-3$ and $3$, while keeping all other units fixed to their inferred values. The participants were asked to describe the resulting transformation of the reconstructed face using one of the 16 label options extracted from the list of 46 descriptive face attributes compiled in consultation with sketch artists in order to produce reliable and identifiable sketches of faces[61]. The full set of 46 attributes was not used, because it would have been too long and confusing for the participants to parse. Furthermore, many of the attributes in the list were not applicable to the current study (e.g. eye colour). To select the subset of attributes to use in this study, we recruited 6 pilot participants and asked them to match the traversals to the most appropriate descriptors selected from the full list of 46 attributes, and used the union of these as the final set. We also re-worded some of the attribute names to make them more appropriate for labelling transformations (e.g. we changed "small eyes" to "eye size"). The participants were presented with three worked examples before being asked to make the judgements. The presentation order of the latent traversals to be judged was randomised.

The experiment took on average $670 \pm 430$ s to complete. For each latent unit, we calculated the distribution of labels chosen by the participants over all labelling options. To measure how semantically meaningful each latent was, we

calculated the entropy of the resulting distribution (being further away from the maximum entropy indicates better consensus by the participants on the semantic meaning of the latent), as well as the maximum proportion of participants who agreed on the label for each latent. We found that 6/11 latents had an agreement of >30% (significantly above the 11.82% agreement expected by chance, $p = 0.001$), with the observed entropy unlikely under the uniform prior distribution ($p = 0.0001$, see Supplementary Fig. 2). Note that for some of the latent units, we found that the participants' label choices were split between closely related concepts, e.g. eye distance and eye size; hair density and hair length; face length, face shape and forehead size; gender and hair length; and age and hair density.

**Human psychophysics: comparing face reconstructions.** We measured the subjective quality of VAE, AE and $\beta$-VAE reconstruction of novel faces from single neurons by asking 300 participants to rank the three reconstructions as "Best", "OK" or "Worst" using a randomised block design. Out of the 62 novel faces used in this study, we had to remove 5 due to legal requirements. The participants were presented with the following instructions: "We would like to compare the quality of face image reconstructions produced by three different systems. You will be presented with 57 face images, each one with three reconstructions by three different systems. The order in which the reconstructions are presented is random, so its position (left, middle or right) does not indicate which system it came from. For each image, we would like you to rate how much the three reconstructions resemble the original. We recommend that you make your judgement on the first holistic impression.".

The experiment took on average $1070 \pm 725$ s to complete. Friedman test[62] with $\alpha = 0.05$ significance level (Python 3.6 scipy stats.friedmanchisquare implementation) was applied to the collected ranking scores, and the null hypothesis that all three models were the same in terms of their reconstruction quality was rejected. Post hoc pairwise comparisons across all 57 images revealed no statistically significant difference between $\beta$-VAE and VAE reconstructions, while significant differences were found between AE and the other two models. Post hoc pairwise comparisons for each image are presented in Supplementary Fig. 3.

**Variance explained.** We used Lasso regression to predict the response of each neuron $\mathbf{n}_j$ from model units. We used 10-fold cross-validation using standardised units and neural responses to find the sparsest weight matrix that produced mean squared error (MSE) results between the predicted neural responses $\hat{\mathbf{n}}_j$ and the real neural responses $\mathbf{n}_j$ no more than one standard error away from the smallest MSE obtained using 100 lambda values. The learnt weight vectors were used to predict the neural responses from model units on the test set of images. Variance explained (VE) was calculated on the test set according to the following:

$$VE_j = 1 - \frac{\sum_i (\hat{\mathbf{n}}_{ij} - \mathbf{n}_{ij})^2}{\sum_i (\mathbf{n}_{ij} - \overline{\mathbf{n}}_j)^2} \qquad (5)$$

where $j$ is the neuron index, $i$ is the test image index, and $\overline{\mathbf{n}}_j$ is the mean response magnitude for neuron $j$ across all test images. In order to speed up the Lasso regression calculations, we manually zeroed out the responses of those model units that did not carry much information about the face images. We defined units as "uninformative" if their standardised responses had low variance $\sigma^2 < 0.01$ across the dataset of 2100 faces. We verified that this did not affect the sparsity of the resulting Lasso regression weights.

Note that the proportion of explained variance for each neuron is typically normalised by the neuron's Spearman-Brown corrected split-half self-consistency over image presentation repetitions (e.g. see Yamins et al.[10]). This data, however, was not available to us, hence our encoding-explained variance results are artificially lower than the typically reported values due to noise. Indeed, Chang et al.[6] used a different classification-based method to quantify how well $\beta$-VAE, AAM and VGG can explain the responses of the same 159 neurons. Their approach did not require noise normalisation, and the results reported show strong performance across models. Our variance explained that the results are consistent with those reported in Chang et al.[6].

**Alignment score.** Two versions of the same measure were simultaneously and independently proposed in the machine learning literature, referred to as "completeness"[29] or "compactness"[63]. We refer to the same measure as "alignment" for more intuitive exposition in this work.

This score measures how well the responses of single neurons are explained by single model units. Perfect score of 1 is achieved if each neuron that is well explained, is only explained well by the activity of a single model unit. If more than one model unit is necessary to explain the activity of a single neuron, this score is reduced. However, if the activity of a neuron cannot be explained at all by any of the model units, the score is unaffected.

First, we obtained the matrix $R$ necessary for calculating the score by training Lasso regressors to predict the responses of each neuron from the population of model latent units. When calculating completeness against the original neural responses, we followed the same procedure as per the variance explained

calculations. When calculating completeness against the artificial (linearly recombined) neural responses, we did not zero out the responses of the "uninformative" units, since in this case this procedure affected the sparsity of the resulting Lasso regression weights. Instead, in order to speed up calculations, we reduced the number of cross-validation splits from ten to three. The completeness score $C_j$ for neuron $j$ was calculated according to the following:

$$C_j = \rho_j(1 - H(p_j)) \quad (6)$$

$$H(p_j) = -\sum_d p_{dj} \log_D p_{dj} \quad (7)$$

$$p_{dj} = \frac{R_{dj}}{\sum_d R_{dj}} \quad (8)$$

$$\rho_j = \frac{\sum_d R_{dj}}{\sum_{dj} R_{dj}} \quad (9)$$

where $j$ indexes over neurons, $d$ indexes over model units, and $D$ is the total number of model units. The overall completeness score per model is equal to the sum of all per-neuron completeness scores $C = \sum_j C_j$. See ref. [29] for more details.

**Unsupervised Disentanglement Ranking (UDR) score.** The UDR score[31] measures the quality of disentanglement achieved by trained $\beta$-VAE models by performing pairwise comparisons between the representations learnt by models trained using the same hyperparameter setting but with different seeds. The measure relies on the assumption that for any particular dataset, well-disentangled $\beta$-VAE models will converge on the same representation up to permutation, subsetting (different models may discover subsets of all disentangled dimensions), and sign inverse (for example, some models may learn to represent age from young to old, while others may represent it from old to young). This approach requires no access to labels or neural data. We used the Spearman version of the UDR score described in ref. [31]. For each trained $\beta$-VAE model, we performed 9 pairwise comparisons with all other models trained with the same $\beta$ value and calculated the corresponding $UDR_{ij}$ score, where $i$ and $j$ index the two $\beta$-VAE models. Each $UDR_{ij}$ score is calculated by computing the similarity matrix $R_{ij}$, where each entry is the Spearman correlation between the responses of individual latent units of the two models. The absolute value of the similarity matrix is then taken $|R_{ij}|$ and the final score for each pair of models is calculated according to

$$\frac{1}{d_a + d_b}\left[\sum_b \frac{r_a^2 * I_{KL}(b)}{\sum_a R(a,b)} + \sum_a \frac{r_b^2 * I_{KL}(a)}{\sum_b R(a,b)}\right] \quad (10)$$

where $a$ and $b$ index into the latent units of models $i$ and $j$, respectively, $r_a = \max_a R(a,b)$ and $r_b = \max_b R(a,b)$. $I_{KL}$ indicate the "informative" latent units within each model, and $d$ is the number of such latent units. The final score for model $i$ is calculated by taking the median of $UDR_{ij}$ across all $j$.

**Average correlation ratio and average unit proportion.** For each neuron, we calculated the absolute magnitude of Pearson correlation with each of the "informative" model units. We then calculated the ratio between the highest correlation and the sum of all correlations per neuron. The ratio scores were then averaged (mean) across the set of unique model units with the highest ratios, and this formed the average correlation ratio score per model. The number of unique model units with the highest ratios divided by the total number of informative model units formed the average unit proportion score.

**Decoding novel faces from single neurons.** We first found the best one-to-one match between single model units and corresponding single neurons. To do this, we calculated a correlation matrix $D_{ij} = \text{Corr}(\mathbf{z}_i, \mathbf{r}_j)$ between the responses of each model unit $\mathbf{z}_i$ and the responses of each neuron $\mathbf{r}_j$ over the subset of 2100 face images that were seen by both the models and the primates, where Corr stands for Pearson correlation. We then used MATLAB R2017b implementation of Kuhn-Munkres[64,65] algorithm (matchpairs) to find the best one-to-one assignment between each model unit and a unique neuron based on the lowest overall $(1 - D_{ij})$ score across all matchings. We used the resulting one-to-one assignments to regress the responses of single latent units from the responses of their corresponding single neurons to the held out 62 faces, using the same subset of 2100 face images that were seen by both the models and the primates for estimating the regression parameters. We standardised both model units and neural responses for the regression. The resulting predicted latent unit responses were fed into the pre-trained model decoder to obtain reconstructions of the novel faces. We calculated the cosine distance between the standardised predicted and real latent unit responses for each face (after filtering out the "uninformative" units), and presented the mean scores across the 62 held out faces for each model.

**Statistical tests.** We used a MATLAB R2017b implementation of Welsch's $t$-test (ttest2) for all pairwise model comparisons, with unequal variance and $\alpha = 0.01$.

**Reporting summary.** Further information on research design is available in the Nature Research Reporting Summary linked to this article.

## Data availability
The unprocessed responses of all models to the 2162 face images generated in this study have been deposited in the figshare database (https://doi.org/10.6084/m9.figshare.c.5613197.v2). This includes AAM, VGG (PCA), VAE and $\beta$-VAE responses previously published in Chang et al.[6]. The figshare database also includes the anonymised psychophysics data, a file describing how the semantic labels used in one of the psychophysics study were obtained from the larger list of 46 descriptive face attributes compiled in Klare et al.[61], and the two sample forms used for data collection on Prolific. The raw neural data supporting the current study were previously published in Chang et al.[6] and are available under restricted access because of the complexity of the customised data structure and the size of the data; access can be obtained by contacting Le Chang (stevenlechang@gmail.com) or Doris Tsao (tsao.doris@gmail.com). The face image data used in this study are available in the corresponding databases: FERET face database[55] (https://www.nist.gov/itl/iad/image-group/color-feret-database), CVL face database[54] (http://lrv.fri.uni-lj.si/facedb.html), MR2 face database[56] (http://ninastrohminger.com/the-mr2), PEAL face database[57], AR face database[51] (http://www2.ece.ohio-state.edu/aleix/ARdatabase.html), Chicago face database[53] (https://www.chicagofaces.org) and CelebA face database[52] (http://mmlab.ie.cuhk.edu.hk/projects/CelebA.html). Source data are provided with this paper.

## Code availability
The code that supports the findings of this study is available upon request from Irina Higgins (irinah@google.com) due to its complexity and partial reliance on proprietary libraries. Open-source implementations of the $\beta$-VAE model, the alignment score and the UDR measure are available at https://github.com/google-research/disentanglement_lib.

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

## Acknowledgements

We would like to thank Raia Hadsell, Zeb Kurth-Nelson and Koray Kavukcuoglu for comments on the manuscript, and Lucy Campbell-Gillingham and Kevin McKee for their help in running the human psychophysics experiments. Funding source: NIH (DP1-NS083063, R01-EY030650) and the Howard Hughes Medical Institute.

## Author contributions

C.S., D.T. and M.B. contributed equally to this manuscript. Conceptualisation, I.H., D.H., M.B.; Methodology, Software, Data Curation and Validation, I.H. and L.C.; Formal Analysis, Investigation, Visualisation, Writing - Original Draft, I.H.; Writing - Review and Editing, I.H., C.S., L.C., D.T., M.B. and C.S.; Project Administration, V.L.; Supervision, D.H., C.S., D.T. and M.B.; Funding Acquisition and Resources, D.T., M.B. and D.H.

## Competing interests

The authors declare no competing interests.

**Additional information**

