## [Peer Review File. · Nature Communications]

Unsupervised deep learning identifies semantic disentanglement in single inferotemporal face patch neuronsEditorial Note: This manuscript has been previously reviewed at another journal that is not operating a transparent peer review scheme. This document only contains reviewer comments and rebuttal letters for versions considered at *Nature Communications*.

REVIEWER COMMENTS

Reviewer #1 (Remarks to the Author):

I was a reviewer for the original version of this manuscript. At that time, I had two major concerns – the framing of the study and the claim that the beta-VAE produces interpretable latent variables, which given the alignment between the neural and model data suggests that single neurons encode interpretable dimensions too.

The reviewers have almost completely addressed my concerns about the framing (but see my comments about the abstract below). The contribution and logic behind the work is now much clearer and the manuscript stronger because of it.

However, I have strong reservations about some of the human psychophysics data presented. While this may seem to be a small part of the manuscript, I think it is critical because the authors attach so much weight to the idea that the latent units are interpretable.

a) Abstract

Despite the fact that all three reviewers questioned the claim that the results using faces might generalize to all objects, and the fact the authors even stated they were weakening the claim in their responses, the abstract still opens with a very broad statement about the ventral visual pathway and the suggestion that objects might be a “microcosm of object recognition”. I’m not sure why the authors are so insistent on promoting this contentious claim, because it tends to detract from what they have actually shown. I think the authors should remove the first two sentences of the abstract and simply acknowledge they are focused on faces. In the discussion they bring up the potential for the results to generalize beyond faces and I think this is a fine point to bring up in the discussion, although I can imagine that many will disagree.

The authors mention the “handcrafted “gold standard” model of face perception”, but within the abstract it’s not all clear what is being referred to.

b) Human psychophysics

I am very happy to see the authors collected psychophysics data to test the interpretability of the latent units, but there are aspects of the approach that are unclear, and some questionable analytic procedures.

- 1) How were the subset of attributes selected from the list of 46?
- 2) What were the 16 attributes?
- 3) Given the a priori selection of labels, it is not appropriate to post-hoc combine labels, especially independently for different latent units (which is what it seems the authors have done). This will artificially inflate the apparent agreement across participants. If a participant chose hair length when gender was also an option, this means they thought hair length was a better descriptor.
- 4) If the latent units are easily interpretable, then asking participants to freely label them would be a better measure. This would also allow participants multiple features of faces that might be varying.

I understand the authors might not want to collect additional data (e.g. point 4 above), but at the very least they should present the data without post-hoc combining labels.

Part of the problem is that the latent units appear to reflect multiple possible dimensions in faces in

complex ways, and not just ones that covary. Looking at the figures, my own sense is that the authors are overstating the extent to which the latent units are easily interpretable. If they want to push the interpretability point further, they should provide stronger data supporting this claim.

c) Figure 2b

Are the lower plots mis-labelled? In the top row the unit numbers correspond to the bar with the highest variance explained, but this is not true for the lower plots. Further, the lower right plot is labelled 'unit 11', but there are only 11 units listed on the x-axis. Or am I misinterpreting what is plotted here?

Reviewer #3 (Remarks to the Author):

I thank the reviewers for their rebuttal. My evaluation of their responses and the changes effected to the MS are mixed.

On the positive side, the authors have toned down the claims such that their conclusions and interpretations are likely warranted by the data. They have also validated their claims on the interpretability of the latent variables sufficiently.

On the negative side I see two main issues. For one, reading Reviewer 2's comments convinced me that out-of-sample generalisability needs to be addressed. While I would want to avoid raising novel points in a revision, this is not a novel point in so far as that R2 raised it already. The authors do not address this concern at all in my opinion (Figure S5 does not even have units). Given that R2 made several constructive suggestions of how to do this, I expect the authors to address this issue with additional analyses.

The second issue is how we weigh different properties of models such as how much variance they explain vs. what theoretical constraint they fulfil. In particular. The variance explained by beta-VAE is low, and lower than AAM when data dimensionality goes up. Reconstruction performance is not better for beta-VE than for AE. At the very least I expect the following:

a) The authors do not hide the inferior predictive power of the beta-VAE model (157-162). They should address this heads-on and in the main MS, not in the supplementary material. Discuss that at higher dimensionality AAM and other models predicted better. Please clarify or hypothesise why? If there is a purely redundant code in the brain (as the authors claim 87-94) I would not see this happening?

b) The authors should make explicit their modelling goals in relation to previous studies' modelling goals in the main MS. One goal is to explain variance in the data. Here the authors fail to provide interesting results. Another goal might be to provide a model given particular constraints (such as low-dim interpretable latent variables). The reader must be informed directly what the goals are, and what the goals were not.

We must appreciate the multitude of goals in computational modelling. But I expect the authors to be as clear as possible on these issues. There is much amiss in blindly fitting models with the only goal to explain most variance, but fitting toy models that exhibit some or other desired characteristic but do not predict is not the preferred way forward either.

It is a choice. Either the manuscript is clear, honest, making an interesting and well circumscribed contribution whose value is unclear, but there is potential. Then it receives the benefit of the doubt. Or it is unclear what its goals exactly are while its performance on key metrics of the field is low, and then it would not be of interest to a broad audience.

Reviewer #1:

I was a reviewer for the original version of this manuscript. At that time, I had two major concerns – the framing of the study and the claim that the beta-VAE produces interpretable latent variables, which given the alignment between the neural and model data suggests that single neurons encode interpretable dimensions too.

The reviewers have almost completely addressed my concerns about the framing (but see my comments about the abstract below). The contribution and logic behind the work is now much clearer and the manuscript stronger because of it.

However, I have strong reservations about some of the human psychophysics data presented. While this may seem to be a small part of the manuscript, I think it is critical because the authors attach so much weight to the idea that the latent units are interpretable.

a) Abstract

Despite the fact that all three reviewers questioned the claim that the results using faces might generalize to all objects, and the fact the authors even stated they were weakening the claim in their responses, the abstract still opens with a very broad statement about the ventral visual pathway and the suggestion that objects might be a “microcosm of object recognition”. I’m not sure why the authors are so insistent on promoting this contentious claim, because it tends to detract from what they have actually shown. I think the authors should remove the first two sentences of the abstract and simply acknowledge they are focused on faces. In the discussion they bring up the potential for the results to generalize beyond faces and I think this is a fine point to bring up in the discussion, although I can imagine that many will disagree.

Thank you for your suggestion. We have removed the first two sentences in the abstract and replaced them with an alternative intro sentence focused on faces.

The authors mention the “handcrafted “gold standard” model of face perception”, but within the abstract it’s not all clear what is being referred to.

We have now made the reference to AMM explicit in the abstract.

b) Human psychophysics

I am very happy to see the authors collected psychophysics data to test the interpretability of the latent units, but there are aspects of the approach that are unclear, and some questionable analytic procedures.

1) How were the subset of attributes selected from the list of 46?

We excluded some attributes because they were not applicable (e.g. there was no colour information in the images because they were greyscale, also none of the people had facial hair or glasses). We then asked 6 pilot participants to select the attributes that they thought described the disentangled features from the remaining subset. The union of these chosen attributes (some were reworded to be more understandable in the context of current study, e.g. changing “small eyes” to “eye size”, since we asked to label transformations) was used as the final set of options presented to the larger participant cohort. See shorturl.at/egFGV for the full list of 46 attributes and a description of whether it was used in the final study, and the reason for rejection if appropriate.

2) What were the 16 attributes?

The following label options were presented to the participants (this was stated in the supplementary materials, but is now also present in the main paper text in the methods section): age, chin size, ethnicity, eye distance, eye size, eye slant, eyebrow position, face length, face shape, forehead size, fringe/bangs, gender, hair density, hair length, nose size, smiling, none of the above.

3) Given the a priori selection of labels, it is not appropriate to post-hoc combine labels, especially independently for different latent units (which is what it seems the authors have done). This will artificially inflate the apparent agreement across participants. If a participant chose hair length when gender was also an option, this means they thought hair length was a better descriptor.

Thank you for spotting this. We have now presented the results without this post-hoc combination step. The results change only slightly:

Old: On average across all 11 units discovered by the beta-VAE 43.3% of the participants agreed on a single distinct semantic label per unit when presented with a choice of 17 options including “none of the above” (significantly above the 15.40% chance level, $p < 0.01$; minimum agreement per unit 20.5%, maximum agreement per unit 76.2%). 8/11 latents had agreement of $> 30\%$ with the observed entropy unlikely under the uniform prior distribution ($p < 0.0001$).

Figure S3: **Latent traversal label agreement between human participants.** Number of participants choosing particular label options from 13 possibilities after semantically similar label choices were consolidated. Numbers in brackets indicate entropy in nats and maximum rater agreement as percentage of the 300 participants. Maximum entropy - 2.56.

New: On average across all 11 units discovered by the beta-VAE 32.1% of the participants agreed on a single distinct semantic label per unit when presented with a choice of 17 options including "none of the above" (agreement significantly above the 11.82% chance level, $p < 0.01$; minimum agreement per unit 11.2%, maximum agreement per unit 70.6%). 6/11 latents had agreement of $>30\%$ with the observed entropy unlikely under the uniform prior distribution ($p < 0.0001$).

Figure S3: **Latent traversal label agreement between human participants.** Number of participants choosing particular label options from 17 possibilities. Numbers in brackets indicate entropy in nats and maximum rater agreement as percentage of the 300 participants. Maximum entropy - 2.83.

4) If the latent units are easily interpretable, then asking participants to freely label them would be a better measure. This would also allow participants multiple features of faces that might be varying.

Thank you for this suggestion. We did consider the free labeling option, but we felt that it would be much harder to quantify the collected data. We think likely that the results would depend heavily on the way we chose to aggregate the freely labelled responses (or the way we instructed raters to do so).

I understand the authors might not want to collect additional data (e.g. point 4 above), but at the very least they should present the data without post-hoc combining labels.

Part of the problem is that the latent units appear to reflect multiple possible dimensions in faces in complex ways, and not just ones that covary. Looking at the figures, my own sense is that the authors are overstating the extent to which the latent units are easily interpretable. If they want to push the interpretability point further, they should provide stronger data supporting this claim.

We agree that there is no readily available metric via which we can quantify the interpretability of the latent units. This is a really fundamental issue because of course there is no consensus over the set of dimensions that define face space in general, and much less for the specific images in the dataset to which we had access (which contains only 2,100 faces). We note that in our previous work using larger face or object datasets, the mapping between latent units and semantically meaningful variables is much stronger and subjectively noticeable. The psychophysical study, which was suggested in an earlier round of reviews, was specifically geared towards testing against the null hypothesis that people cannot agree upon the variable coded by each latent unit - a hypothesis that is roundly rejected by our analyses. Nevertheless, we accept the caveats raised by the reviewer and have tempered our claims (e.g. adding modifiers like “fairly” or “largely” in front of “semantically interpretable”), and explicitly flagging the challenges in providing evidence for the claim that single latent units encode semantically meaningful variables.

c) Figure 2b

Are the lower plots mis-labelled? In the top row the unit numbers correspond to the bar with the highest variance explained, but this is not true for the lower plots. Further, the lower right plot is labelled ‘unit 11’, but there are only 11 units listed on the x-axis. Or am I misinterpreting what is plotted here?

Thank you for spotting this typo. We introduced a largely arbitrary ordering for the latent dimensions for our human psychophysics experiment, and have accidentally used the ordering from that study to label the latents at the top of the bar plots, while the bar plots themselves corresponded to the particular latent ordering in one of the disentangled beta-VAEs used for the psychophysics experiment. We have now updated the plot.

Reviewer #3:

I thank the reviewers for their rebuttal. My evaluation of their responses and the changes effected to the MS are mixed.

On the positive side, the authors have toned down the claims such that their conclusions and interpretations are likely warranted by the data. They have also validated their claims on the interpretability of the latent variables sufficiently.

On the negative side I see two main issues. For one, reading Reviewer 2's comments convinced me that out-of-sample generalisability needs to be addressed. While I would want to avoid raising novel points in a revision, this is not a novel point in so far as R2 raised it already. The authors do not address this concern at all in my opinion (Figure S5 does not even have units). Given that R2 made several constructive suggestions of how to do this, I expect the authors to address this issue with additional analyses.

We assume that the reviewer is referring to the following request by R2 from the previous review round: "One quite easy to imagine possibility would be to train a β -VAE on 2100 other face images, and then make the comparison on held-out images. This would also measure generalization in an effective way, which is kind of missing (I think) in the current setup."

In the specific context of our paper, our understanding of this request is the following. By asking us to train beta-VAE on a dataset of faces other than the 2,100 used for testing, the reviewers would like us to check whether beta-VAE has learnt general features useful for describing the faces, or whether the model has instead discovered something idiosyncratic about the 2,100 test faces by overfitting to the training data distribution. Past work has verified that beta-VAE consistently discovers stable and generally useful factors that span the space of the dataset generative factors without overfitting (e.g. see Higgins et al, 2017; Locatello et al, 2018; Lee et al, 2020; Watters et al, 2019). When it comes to the particular dataset of 2,100 faces used in this study, we feel confident that beta-VAE has once again learnt generally useful features that span the space of faces because naive human participants were able achieve a high degree of consistency in assigning terms deemed to be informative for describing faces to sketch artists to the individual dimensions of the beta-VAE during our psychophysical study.

The dataset of faces used in this paper already contains all the face images suitable for this study - forward-facing, unobstructed faces in neutral lighting conditions, without strong emotional expressions etc. - from across all the publicly available datasets of faces as described in Chang et al, 2021. For this reason we have been unable to find another large-scale dataset of faces with similar distributional statistics to the test set of 2,100 faces in order to train our models and test the out-of-sample generalisability of our findings. Nevertheless, to address the reviewers' comments, we have extended our analysis comparing the train and test data distributions used in this work. We have been able to show that the dataset used for training the

models, which we produced by applying pre-processing augmentations to the dataset of 2,100 original faces as discussed in the methods section, is seen by the models as significantly different from the original 2,100 faces used to compare the models and the neural responses. Not only that, we are also able to show that this difference is as large as the difference between the original images and a set of 62 held out novel faces that the models never saw during training (see updated Fig. S5a). As mentioned above, our understanding of the reviewers' comments is that they would like to see whether beta-VAE can learn generally useful dimensions for representing faces that match those discovered by the neurons in the brain on a dataset that is significantly different from the test images - results that suggest that beta-VAE does not just memorise something idiosyncratic about the test faces themselves (does not overfit to them). We believe our new Fig. S5a suggests that this is indeed the case, and that the results reported in our paper are already equivalent to the out-of-sample generalisation setting requested by the reviewers.

Figure S5: **Alignment results broadly generalise to heldout faces.** **a.** Circles, Negative Log Likelihood (NLL) averaged per pixel for 2,100 face images under the learnt model distribution. Lower is better, with the perfect model scoring 0. Results are presented for the model instances shown in Fig. 6c. Train, same mirror-flipped 2,100 faces used for training the models. Test, original 2,100 faces that were presented to the primates. Heldout, novel 62 faces not seen during training. No significant difference is found between test and heldout NLL results (all $p > 0.05$, Welsch's t-test); train and test, as well as train and heldout NLL results are significantly different across all models (all $p < 0.01$, Welsch's t-test). β -VAE train NLL results are significantly higher than AE or VAE train NLL results (all $p < 0.01$, Welsch's t-test). **b.** Alignment scores calculated over 62 heldout faces. No significant difference is found between the alignment scores of neuron subsets, β -VAE and VGG (raw) (all $p > 0.05$, Welsch's t-test). Boxplot details same as in Fig. 3. Model numbers same as in Figs. 2b and 3a apart from VGG (raw), $n=23$; Classifier, $n=71$.

To further validate this, however, we re-calculated the alignment, average correlation ratio and average unit proportion scores that make up the main results of the paper using the 62 held out faces coming from the same data distribution as the 2,100 faces presented to the primates, but which none of the models saw during training. The results are slightly different due to the inherent noise that comes from performing any calculations in such a low data regime, combined with the low signal-to-noise ratio of the neural data (Fig. S6a illustrates the high signal-to-noise ratio problem in the low data regime). Saying this, Figs. S5b and S6b-c show that our main results broadly hold even in this out-of-sample generalisation setting: beta-VAE still has the highest scores across the board. We have added an additional section in the Supplementary Materials with a more detailed discussion of these results.

Figure S6: **One-to-one correlation and diversity results broadly generalise to heldout faces.**

a. Standard deviation of absolute Pearson correlation between the responses of single neurons and single model units when calculated using 2,100 test faces or 62 heldout faces. Each dot corresponds to a single model from across all model classes ($n=449$). Correlation scores are significantly more variable in the low-data regime of heldout faces ($p < 0.01$, Welch's t-test). **b.** Average correlation ratio scores. **c.** Average unit proportion scores. Both calculated using heldout 62 face data. Boxplot details same as in Fig. 5.

The second issue is how we weigh different properties of models such as how much variance they explain vs. what theoretical constraint they fulfil. In particular. The variance explained by beta-VAE is low, and lower than AAM when data dimensionality goes up. Reconstruction performance is not better for beta-VE than for AE. At the very least I expect the following:

a) The authors do not hide the inferior predictive power of the beta-VAE model (157-162). They should address this heads-on and in the main MS, not in the supplementary material. Discuss that at higher dimensionality AAM and other models predicted better. Please clarify or hypothesise why? If there is a purely redundant code in the brain (as the authors claim 87-94) I would not see this happening?

Thank you for the suggestion. We have taken it on board and have moved our variance explained results from the Supplementary Materials to the main text as requested (see new Fig. 7). We have also added a new subsection in the Results of the manuscript to address this point head on for maximal transparency:

“Disentangled units correspond to a subset of all dimensions that span the space of faces. So far we have shown that the disentangled representational form in the beta-VAE is a closer match to the representational form of real neurons compared to the alternatives presented by the baselines. This, however, is not the whole story. A complementary question to ask is to what degree the information captured by the beta-VAE representation overlaps with the information captured by the neural population (see Fig. 7c). Hence, we calculated how much neural response variance was explained by the beta-VAE (encoding variance explained), and how much variance in the model responses was explained by the neurons (decoding variance explained) (see Fig. 7a-b). While the absolute values presented are artificially low due to the lack of noise normalisation, our results are consistent with those reported in Chang et al 2019 -- beta-VAE representations contain less information in general than the other baselines (apart from the Classifier and VGG raw), but the information that does get preserved by the beta-VAE overlaps with the information within the neural population the most compared to the other baselines (apart from AE). Taken together, the results presented in Fig. 7a-b suggest that in terms of linearly decodable information overlap with the neural population, the best models are AE and beta-VAE (closer to the bottom left quadrant in Fig. 7c, while the other baselines are closer to the top left or right quadrants). “

We have also added a paragraph to the discussion of the manuscript:

“It is worth noting that in general the baselines, including AAM, contain a larger number informative dimensions than β -VAE. Furthermore, it has been demonstrated that when the larger full set of AAM dimensions is used, it explains more neural variance at the population level than the smaller full set of disentangled β -VAE units [6]. Indeed, it is clear that the handful of disentangled dimensions discovered by β -VAE in the current study are not sufficient to fully describe the whole space of faces. The reason why β -VAE discovers only a subset of the dimensions necessary to describe faces fully stems from a known limitation of the current

methods for disentangled representation learning -- they rely on training on well aligned large datasets for achieving maximal interpretability and disentangling quality. Indeed, applying β -VAE to a more suitable dataset of faces [43] allows it to recover at least double the number of disentangled dimensions than the number found in the current study (see Supplementary Materials for more details, example latent traversals shown in Fig. 1a, second from the right). The lack of appropriate levels of scale and alignment for the dataset of 2,100 faces used in this study leaves room for improvements both in terms of latent interpretability and the amount of population neural variance explained to future work.

b) The authors should make explicit their modelling goals in relation to previous studies' modelling goals in the main MS. One goal is to explain variance in the data. Here the authors fail to provide interesting results. Another goal might be to provide a model given particular constraints (such as low-dim interpretable latent variables). The reader must be informed directly what the goals are, and what the goals were not.

We must appreciate the multitude of goals in computational modelling. But I expect the authors to be as clear as possible on these issues. There is much amiss in blindly fitting models with the only goal to explain most variance, but fitting toy models that exhibit some or other desired characteristic but do not predict is not the preferred way forward either.

It is a choice. Either the manuscript is clear, honest, making an interesting and well circumscribed contribution whose value is unclear, but there is potential. Then it receives the benefit of the doubt. Or it is unclear what its goals exactly are while its performance on key metrics of the field is low, and then it would not be of interest to a broad audience.

This is a very important point, thank you for stating it so explicitly here. We have modified the paper (introduction and results) to make the goal of the paper clearer:

“The goal of this paper is to answer primarily the first part of this question -- whether a general learning objective can give rise to a representation that matches the representational form employed by the real neurons -- because we believe that this important question has so far been ignored in the literature, with all quantitative results instead reporting measures of explanatory power [3,6,8,11,13-16] that are insensitive to the representational form [17]”

“The alignment score measures whether the representational form within a subset of the neural population is similar to the representational form discovered by the model, which is a different yet complementary goal to other measures commonly used in the literature [8,16], which instead measure the amount of linearly accessible information shared by the neural and model representations at the population level, while being insensitive to the representational form.”

There is also a more extensive discussion of the different goals and measures in the discussion section (second paragraph from the end):

“One contribution of this paper is the introduction of novel measures for comparing neural and model representations. Unlike other often used representation comparison methods (e.g. explained variance of neuron-level regressions [8,11,13-15] or Representational Similarity Analysis (RSA) [11,16]) which are insensitive to invertible linear transformations, our methods measure the alignment between individual neurons and model units. Hence, they do not abstract away the representational form and preserve the ability to discriminate between alternative computational models that may otherwise score similarly [17]. To summarise, while the traditional methods compare the informativeness of representations, our approach compares their representational form, hence the two are complementary to each other.”

REVIEWER COMMENTS

Reviewer #1 (Remarks to the Author):

The authors have addressed my concerns satisfactorily and I have no further comments.

Reviewer #3 (Remarks to the Author):

In this round of revisions the authors have addressed my comments sufficiently. I am ambivalent about the confidence I have in the conclusion drawn, but the authors have convinced me by the additional analyses that they should receive the benefit of the doubt. They also clarified their scientific goals sufficiently for the community to judge their work in the proper context. I have no further reservations and thus recommend the MS for publication.